# DEMYSTIFYING MASKGIT SAMPLER AND BEYOND: ADAPTIVE ORDER SELECTION IN MASKED DIFFUSION

## ABSTRACT

Masked diffusion models have shown promising performance in generating high-quality samples in a wide range of domains, but accelerating their sampling process remains relatively underexplored. To investigate efficient samplers for masked diffusion, this paper theoretically analyzes the MaskGIT sampler for image modeling, revealing its implicit temperature sampling mechanism. Through this analysis, we introduce the "moment sampler," an asymptotically equivalent but more tractable and interpretable alternative to MaskGIT, which employs a "choose-then-sample" approach by selecting unmasking positions before sampling tokens. In addition, we improve the efficiency of choose-then-sample algorithms through two key innovations: a partial caching technique for transformers that approximates longer sampling trajectories without proportional computational cost, and a hybrid approach formalizing the exploration-exploitation trade-off in adaptive unmasking. Experiments in image and text domains demonstrate our theory as well as the efficiency of our proposed methods, advancing both theoretical understanding and practical implementation of masked diffusion samplers.

## 1 INTRODUCTION

Generative models have witnessed remarkable progress in recent years, with diffusion models (Sohl-Dickstein et al., 2015) emerging as a dominant paradigm across various domains including images (Ho et al., 2020; Dhariwal & Nichol, 2021), audio (Kong et al., 2021) and video (Ho et al., 2022). While continuous diffusion models have garnered significant attention, discrete diffusion models (Austin et al., 2021; Campbell et al., 2022; Gu et al., 2022; Lou et al., 2024) offer compelling advantages for inherently discrete data, such as tokenized representations in all the above domains.

Among discrete diffusion approaches, masked diffusion models (Sahoo et al., 2024; Shi et al., 2024; Ou et al., 2025) have demonstrated exceptional performance, particularly in generating high-quality samples in language domains. However, their sampling process remains computationally intensive, requiring hundreds of function evaluations. Recent efforts to accelerate sampling in discrete/masked diffusion models have explored various directions, including optimized scheduling (Park et al., 2025), distillation (Deschenaux & Gulcehre, 2025; Hayakawa et al., 2025; Zhu et al., 2025), and adaptive token selection (Kim et al., 2025; Ben-Hamu et al., 2025).

One of the most notable masked diffusion samplers is given by MaskGIT (Chang et al., 2022), which has shown that impressive image generation can be achieved with as few as 8-12 sampling steps, significantly fewer than the hundreds typically required by discrete diffusion models. This efficiency has inspired subsequent work in image (Lezama et al., 2023; Besnier et al., 2025), audio (Garcia et al., 2023; Comunità et al., 2024), and language domains (Zheng et al., 2024; Nie et al., 2025). However, the method is heuristic and not well understood in theory, and it often shows degraded performance when increasing sampling steps (e.g., Gat et al., 2024; this paper's Figure 3).

In this paper, to understand the behavior of the MaskGIT sampler, we start by providing its theoretical analysis. We reveal that it implicitly performs temperature sampling, which explains the aforementioned degraded performance. Through the analysis, we obtain the *moment sampler*, a more tractable/interpretable sampler which is asymptotically equivalent to MaskGIT. Unlike MaskGIT, the moment sampler chooses the unmasking positions *before* sampling tokens, which we call a *choose-then-sample* strategy, and this transformation (from MaskGIT, which is a "sample-then-choose" strategy) enables us to further improve the method (see Figure 1).

Figure 1: Overview of our contributions. Both samplers determine tokens at two out of five positions, but with different order of positional choice and token sampling. While they are asymptotically equivalent as we show in Theorem 2, the moment sampler belongs to the family of "choose-then-sample" methods, which we can further enhance in two ways as described in Section 4.

For practical improvement of masked diffusion samplers, we then introduce two key techniques specifically tailored for choose-then-sample methods. First, we propose a partial caching technique for transformer-based models that effectively approximates the sampling trajectories with more steps without proportionally increasing computational cost, unlike the MaskGIT sampler that requires re-computation of all positions at each step. Second, we formalize the exploration-exploitation trade-off in adaptive unmasking of masked diffusion sampling, leading to a hybrid approach that combines the strengths of exploitation-focused methods (Kim et al., 2025; Ben-Hamu et al., 2025) with exploration-oriented techniques such as Halton scheduling (Besnier et al., 2025).

Our contributions are illustrated in Figure 1. They can be summarized as follows:

- In Section 3, we demonstrate that MaskGIT implicitly performs temperature sampling by providing its theoretical analysis (Theorem 2). This finding is essential in understanding MaskGIT, which has been regarded as clever index selection rather than temperature sampling, and it also explains the degraded performance of MaskGIT when the number of steps is large. The analysis is done by introducing the moment sampler, a choose-then-sample style algorithm approximating MaskGIT.

- In Section 4, we propose two techniques for improving choose-then-sample algorithms including moment sampler. One is a partial caching technique for transformer-based models that approximates more steps without proportionally increasing computational cost. The other is through our formalization of the exploration-exploitation trade-off in masked diffusion sampling, developing a hybrid approach that balances these competing objectives.

- In Section 5, we validate our theoretical findings and the efficiency of our proposed methods through experiments in image and text on the ImageNet and OpenWebText datasets.

Our work not only advances the theoretical understanding of masked diffusion samplers but also provides practical techniques to enhance their efficiency, toward better modeling for discrete tokens.

## 2 PRELIMINARIES

Let $q_{\text{data}}$ be the data distribution over the product space $\mathcal{X} = \mathcal{S}^D$, where $\mathcal{S}$ is a finite set of tokens and $D$ is the dimensionality/length of the data. Each state $\boldsymbol{x} \in \mathcal{X}$ can be represented as a sequence of tokens $\boldsymbol{x} = (x_i)_{i=1}^D$ with $x_i \in \mathcal{S}$ for each $i \in [D]$, where $[D] := \{1, \ldots, D\}$. For any set of indices $I \subset [D]$, let us write $\boldsymbol{x}_I = (x_i)_{i \in I}$. For $I, J \subset [D]$, let $q_{I|J}(\boldsymbol{x}_I|\boldsymbol{x}_J) = \mathbb{P}(\boldsymbol{y}_I = \boldsymbol{x}_I \mid \boldsymbol{y}_J = \boldsymbol{x}_J)$ with $\boldsymbol{y} \sim q_{\text{data}}$. Let us also slightly abuse the notation to represent the case $I = \{i\}$ to write $q_{i|J}(x_i|\boldsymbol{x}_J)$. Finally, $\operatorname{argtop} k_{i \in I}\{a_i\}$ is the top-$k$ indices of a sequence $(a_i)_{i \in I}$ rearranged in descending order.

### 2.1 MASKED DIFFUSION MODELS

In masked diffusion models, we augment the vocabulary with a special mask token $\mathtt{M} \notin \mathcal{S}$. The forward process $\boldsymbol{x}(t) \in (\mathcal{S} \cup \{\mathtt{M}\})^D$ with $\mathcal{S}^D \ni \boldsymbol{x}(0) = \boldsymbol{x} \sim q_{\text{data}}$ gradually replaces tokens with mask tokens according to a predefined schedule, eventually reaching $\boldsymbol{x}(1) = (\mathtt{M}, \ldots, \mathtt{M})$. For

generation, we simulate its backward process starting from a fully masked sequence. While the time-dependent formulation is a natural consequence of general discrete diffusion modeling, recent studies (Ou et al., 2025; Zheng et al., 2025) pointed out that, under commonly used forward masking models, the conditional probability $\mathbb{P}(\boldsymbol{x}(0) = \boldsymbol{z} \mid \boldsymbol{x}(t) = \boldsymbol{y})$ does not depend on $t$. Thus, for simplicity, we assume that a given pretrained model is time-independent and tries to approximate the conditional distribution of $q_{I|J}(\boldsymbol{x}_I|\boldsymbol{x}_J)$ for $I, J \subset [D]$, and $\boldsymbol{x} \sim q_{\mathrm{data}}$. Then, our unmasking process is given by $\emptyset = J_0 \subset \cdots \subset J_n = [D]$ to iteratively sample $\boldsymbol{x}_{J_\ell \setminus J_{\ell-1}}$ approximately from $q_{J_\ell \setminus J_{\ell-1} | J_{\ell-1}}(\cdot | \boldsymbol{x}_{J_{\ell-1}})$ for $\ell = 1, \ldots, n$. For efficient sampling, we prefer using smaller $n$, the number of unmasking steps, which requires unmasking of multiple token positions at each step.

Since the probability distributions over $\mathcal{S}^{|I|}$ with large $|\mathcal{S}|$ and/or $|I|$ are intractable, directly modeling $q_{I|J}$ for $I$ containing multiple token positions is inefficient. A common workaround is the *product modeling* $p_{I|J}(\boldsymbol{x}_I|\boldsymbol{x}_J) := \prod_{i \in I} p_{i|J}(x_i|\boldsymbol{x}_J)$, which only approximates one-token marginals as $p_{i|J} \approx q_{i|J}$. In this paper, we investigate efficient samplers for such product models that do not require any retraining or additional components.

## 2.2 MASKGIT SAMPLER

We review MaskGIT, which is a pioneering method of post-hoc efficient sampling for masked diffusion (Chang et al., 2022). To simplify the analysis, let us just consider *one step* of unmasking. Suppose we are given $N$ probability distributions $p_1, \ldots, p_N$ (corresponding to $p_{i|J}$ for $i \notin J$ in the previous section, and so $N = D - |J|$). We would like to unmask $k$ token positions from $[N]$. Given the temperature parameter [1] $\alpha > 0$, the MaskGIT sampler (Chang et al., 2022) is given as follows:

(MG1) Independently sample $x_i \sim p_i$ and a standard Gumbel noise $\xi_i$ for each $i \in [N]$.

(MG2) Choose $(i_1, \ldots, i_k) = \mathrm{argtop}\, k_{i \in [N]} \{\log p_i(x_i) + \alpha \xi_i\}$.

(MG3) Return the indices $i_1, \ldots, i_k$ and samples $x_{i_1}, \ldots, x_{i_k}$.

We formalize this sampling scheme in Algorithm 1 in the appendix. This sampler works surprisingly well in image modeling when the number of sampling steps is small (Chang et al., 2022; Li et al., 2023). However, it is also observed that its performance degrades as we increase the number of steps (Gat et al., 2024; Ren et al., 2025; see also Figure 3). In the following section, we investigate this algorithm and derive an asymptotically equivalent sampler that is more flexible and interpretable; the analysis also partially explains the above performance decay.

## 3 MASKGIT SAMPLER SECRETLY CONDUCTS TEMPERATURE SAMPLING

We analyze the MaskGIT sampler in this section. Let us start from a general result on the Gumbel-top-$k$ sampling, which is used in (MG2) of the MaskGIT sampler. As a top-$k$ generalization of the Gumbel-max trick (Maddison et al., 2014), the following result is known:

**Proposition 1** (Gumbel-top-$k$ trick, Kool et al., 2019, Eq. 18)**.** *Suppose we are given $\mu_1, \ldots, \mu_N \in \mathbb{R}$ and i.i.d. standard Gumbel noise $\xi_1, \ldots, \xi_N$. Let $(i_1^*, \ldots, i_k^*) = \mathrm{argtop}\, k_{i \in [N]} \{\mu_i + \xi_i\}$. Then, for distinct indices $i_1, \ldots, i_\ell \in [N]$ with $\ell \leq k$, we have $\mathbb{P}\big(i_\ell^* = i_\ell \mid i_1^* = i_1, \ldots, i_{\ell-1}^* = i_{\ell-1}\big) = \exp(\mu_{i_\ell}) / \sum_{i \in [N] \setminus I_{\ell-1}} \exp(\mu_i)$, where $I_{\ell-1} := \{i_1, \ldots, i_{\ell-1}\}$.*

As mentioned in the original paper, it is mathematically equivalent to the size-$k$ sampling without replacement with logits $\mu_i$. While it also reveals the value of $\mathbb{P}(i_1^* = i_1, \ldots, i_k^* = i_k)$, we are particularly interested in the conditional form in the proposition.

Let us consider the case $I = [N]$ for simplicity. Let $(i_1^*, \ldots, i_k^*) = \mathrm{argtop}\, k_{i \in [N]} \{\log p_i(x_i) + \alpha \xi_i\}$ in the MaskGIT sampler (Algorithm 1, line 3). Since it is equivalent to consider the Gumbel-top-$k$ sampling for $\mu_i = \alpha^{-1} \log p_i(x_i)$ from Proposition 1, we have

$$\mathbb{P}\big(i_\ell^* = i_\ell \mid i_1^* = i_1, \ldots, i_{\ell-1}^* = i_{\ell-1}, (x_i)_{i=1}^N\big) = \frac{p_{i_\ell}(x_{i_\ell})^{1/\alpha}}{\sum_{i \in [N] \setminus I_{\ell-1}} p_i(x_i)^{1/\alpha}} \tag{1}$$

---

[1] While not mentioned Chang et al. (2022), this "Gumbel temperature" $\alpha$ is used in their official implementation (https://github.com/google-research/maskgit/). Its typical value ranges around 1.0 to 10.0 with additional step-dependent scheduling (Besnier & Chen, 2023; Li et al., 2023; Comunità et al., 2024).

for each $\ell \le k$. When $N - k$ is large (i.e., there are many masked positions), the sum of independent terms $p_i(x_i)^{1/\alpha}$ should be approximated by its expectation because of probability concentration:

$$\sum_{i \in [N] \setminus I_{\ell-1}} p_i(x_i)^{1/\alpha} \approx \sum_{i \in [N] \setminus I_{\ell-1}} \mathbb{E}_{x_i \sim p_i}\left[p_i(x_i)^{1/\alpha}\right] = \sum_{i \in [N] \setminus I_{\ell-1}} \sum_{x \in \mathcal{S}} p_i(x)^{1+1/\alpha}. \tag{2}$$

To see its quantitative version, applying Bernstein's inequality (e.g., Boucheron et al. 2013, Corollary 2.11) to the independent random variables $p_i(x_i)^{1/\alpha} \in [0, 1]$ yields, for $t > 0$,

$$\mathbb{P}\left(\left|\sum_{i \in [N] \setminus I_{\ell-1}} p_i(x_i)^{1/\alpha} - \sum_{i \in [N] \setminus I_{\ell-1}} \mathbb{E}\left[p_i(x_i)^{1/\alpha}\right]\right| \ge t\right) \le 2\exp\left(-\frac{t^2}{2(\sigma^2 + t/3)}\right),$$

where $\sigma^2$ is the variance of $\sum_{i \in [N] \setminus I_{\ell-1}} p_i(x_i)^{1/\alpha}$. Since $0 \le p_i(x_i) \le 1$, it is upper-bounded as $\sigma^2 \le \sum_{i \in [N] \setminus I_{\ell-1}} \mathbb{E}\left[(p_i(x_i)^{1/\alpha})^2\right] \le \sum_{i \in [N] \setminus I_{\ell-1}} \mathbb{E}\left[p_i(x_i)^{1/\alpha}\right]$. By substituting this upper bound and letting $t = \epsilon \sum_{i \in [N] \setminus I_{\ell-1}} \mathbb{E}\left[p_i(x_i)^{1/\alpha}\right]$ for an $\epsilon > 0$ in the Bernstein estimate, we can quantify the concentration of (2) as follows:

$$\mathbb{P}\left(\left|\frac{\sum_{i \in [N] \setminus I_{\ell-1}} p_i(x_i)^{1/\alpha}}{\sum_{i \in [N] \setminus I_{\ell-1}} \mathbb{E}\left[p_i(x_i)^{1/\alpha}\right]} - 1\right| \ge \epsilon\right) \le 2\exp\left(-\frac{3\epsilon}{8} \sum_{i \in [N] \setminus I_{\ell-1}} \mathbb{E}\left[p_i(x_i)^{1/\alpha}\right]\right)$$

$$\le 2\exp\left(-\frac{3(N-k+1)}{8|\mathcal{S}|^{1/\alpha}}\epsilon\right),$$

where the second inequality follows from $N - k + 1 \le N - (\ell - 1)$ and the lower bound of $\mathbb{E}\left[p_i(x_i)^{1/\alpha}\right]$ using the vocabulary size $|\mathcal{S}|$, shown by (17) in the appendix. Although the actual proof is more involved, this concentration is at the heart of proving our main theorem (Theorem 2).

Since $\sum_{x \in \mathcal{S}} p_i(x)^{1+1/\alpha}$ is the $(1 + 1/\alpha)$-th power of $(1 + 1/\alpha)$-norm of $p_i$ when it is regarded as a vector in $\mathbb{R}^{|\mathcal{S}|}$, we can simply write $\sum_{i \in [N] \setminus I_{\ell-1}} p_i(x_i)^{1/\alpha} \approx \sum_{i \in [N] \setminus I_{\ell-1}} \|p_i\|_{1+1/\alpha}^{1+1/\alpha}$. Under this approximation (including the same approximation for smaller $\ell$), we can derive

$$\mathbb{P}\left(x_{i_\ell}, i_\ell^* = i_\ell \mid i_1^* = i_1, \ldots, i_{\ell-1}^* = i_{\ell-1}\right) \approx \frac{p_{i_\ell}(x_{i_\ell})^{1+1/\alpha}}{\sum_{i \in [N] \setminus I_{\ell-1}} \|p_i\|_{1+1/\alpha}^{1+1/\alpha}}, \tag{3}$$

where the right-hand side makes a probability distribution over $\mathcal{S} \times ([N] \setminus I_{\ell-1})$. The derivation of (3) is deferred to Appendix B.1. From Proposition 1, choosing indices $i_1, \ldots, i_\ell$ with the right-hand side of (3) (summed over $x_{i_\ell} \in \mathcal{S}$) is equivalent to Gumbel-top-$k$ sampling with $\mu_i = \log\|p_i\|_{1+1/\alpha}^{1+1/\alpha}$. Then, given the index $i$, the sampling distribution for $x_i$ is proportional to $p_i^{1+1/\alpha}$.

**Moment sampler.** Based on the above analysis, the *moment sampler*, our alternative sampler for approximating MaskGIT, is formulated as follows:

(MM1) Let $(i_1, \ldots, i_k) = \arg\text{top}\,k_{i \in [N]}\{\log\|p_i\|_{1+1/\alpha}^{1+1/\alpha} + \xi_i\}$ with i.i.d standard Gumbel $(\xi_i)_{i=1}^N$.

(MM2) Independently sample $x_i \sim \tilde{p}_i$ with $\tilde{p}_i \propto p_i^{1+1/\alpha}$ for each $i \in \{i_1, \ldots, i_k\}$.

(MM3) Return the indices $i_1, \ldots, i_k$ and samples $x_{i_1}, \ldots, x_{i_k}$.

Since the exponent $1 + 1/\alpha$ in (MM2) is a potential source of sampling error as we discuss later in this section, we also consider the "unbiased" version of second step, i.e., simply sampling from $p_i$. To cover both of these options of sampling from $\propto p_i^\gamma$ with exponent $\gamma = 1 + 1/\alpha$ and 1, we formalize the moment sampler in Algorithm 2 with a general exponent $\gamma$ in (MM2). In the experiments in Section 5, $\gamma = 1 + 1/\alpha$ defines **Moment**, whereas the choice $\gamma = 1$ provides **U-Moment**. See also Figures A and B in the appendix for comparing the MaskGIT and moment samplers in actual implementation using PyTorch.

Mathematically, we can prove the following:

**Theorem 2** (Moment sampler approximates MaskGIT in the $N \gg k^2$ regime)**.** *Let* $p_1, \ldots, p_N$ *be probability distributions over a finite set* $\mathcal{S}$*. For* $k \in [N]$ *and* $\alpha > 0$*, let* $p_{\mathrm{MaskGIT}}$ *be the output distribution of the corresponding MaskGIT sampler (Algorithm 1), i.e., the distribution of* $(i_1, \ldots, i_k)$ *and* $(x_{i_\ell})_{\ell=1}^{k}$ *over* $[N]^k \times \mathcal{S}^k$*. Similarly, let* $p_{\mathrm{moment}}$ *be the output distribution of the moment sampler (Algorithm 2 with* $\gamma = 1 + 1/\alpha$*). Then, we have*

$$d_{\mathrm{TV}}(p_{\mathrm{moment}}, p_{\mathrm{MaskGIT}}) \leq 5 \sqrt{\frac{k^2 |\mathcal{S}|^{1/\alpha}}{N}} \left( 1 + \sqrt{\log^+ \left( \frac{N}{k^2 |\mathcal{S}|^{1/\alpha}} \right)} \right),$$

*where* $d_{\mathrm{TV}}$ *denotes the total variation distance and* $\log^+(x) := \log(\max\{1, x\})$ *for* $x \in \mathbb{R}$*.*

For a formal argument, see Appendix C.6 and Theorem 7. Since $|\mathcal{S}|^{1/\alpha}$ is constant and $N/k$ is approximately the number of steps, it gets tighter when $\#(\text{steps}) \gg \#(\text{unmasked tokens per step})$. In this regime, MaskGIT is approximated well by the moment sampler, which samples from a different (or biased) distribution compared to the true marginal because of the exponent $1 + 1/\alpha$ in (MM2). It thus partially explains the degraded behavior of the MaskGIT sampler as we increase the number of steps. Moreover, even outside the above regime, we empirically observe that the moment sampler shows fairly close behavior with MaskGIT in terms of evaluation metrics (Figures 3 and 5(Left)).

From Theorem 2, we can approximately decompose the MaskGIT sampler into index selection and temperature sampling (Figure 1). This allows better understanding: for instance, in our experiments, temperature sampling is the dominant factor as we can mostly replicate the performance of MaskGIT even if we omit the index selection of moment sampler (Section 5). Also, from a practical viewpoint, we can improve the moment sampler by techniques not applicable to MaskGIT (Section 4).

**Choose-then-sample algorithms.** Arguably the most notable difference of the moment sampler from MaskGIT is that we choose the indices to unmask *before* sampling a token in each position. Let us call this strategy *choose-then-sample* (CTS) in general, which we formalize and analyze below.

Given the set of currently unmasked indices $I$ and sample $\boldsymbol{x}_I$, a step of CTS algorithms over $\mathcal{S}^D$ with inverse temperature $\gamma > 0$ can be formalized as follows:

(CTS1) Sample $J \subset [D] \setminus I$ for where to unmask. Its distribution is denoted as $\pi(\cdot | I, \boldsymbol{x}_I)$.

(CTS2) $x_j \sim p_j$ with $p_j \propto p_{j|I}(\cdot | \boldsymbol{x}_I)^\gamma$ for each $j \in J$.

We also present the whole algorithm as pseudocode in Algorithm 3 in the appendix. The unmasking position distribution $\pi$ is usually determined from the information regarding $(p_{j|I}(\cdot | \boldsymbol{x}_I))_{j \in [D] \setminus I}$ such as entropy and probability margin (Xiang et al., 2023; Kim et al., 2025).

It is worth noting that, unlike the "sample-then-choose" MaskGIT, CTS algorithms without temperature (i.e., $\gamma = 1$) return the correct distribution as we increase the number of steps, provided that its marginals are accurate. We formally prove this in Proposition 3. It is also mentioned by Ben-Hamu et al. (2025, Section 5) without a formal proof, and we prove it for completeness in Appendix C.1.

**Proposition 3** (One-by-one CTS algorithm is unbiased)**.** *In a CTS algorithm with* $\gamma = 1$*, let us further assume that* $J \sim \pi(\cdot | I, \boldsymbol{x}_I)$ *in (CTS1) is always a singleton set (i.e.,* $|J| = 1$*) and* $p_{i|I} = q_{i|I}$ *holds for each* $I \subset [D]$ *and* $i \in [D] \setminus I$*. Then, we have* $\boldsymbol{x} \sim q_{\mathrm{data}}$ *for the generated sample* $\boldsymbol{x}$*.*

## 4    TOWARDS MORE EFFICIENT CHOOSE-THEN-SAMPLE ALGORITHMS

One benefit of CTS algorithms pointed out by Zheng et al. (2025) is that we can avoid sampling from $N$ categorical distributions over $\mathcal{S}$, which can be costly if $N$ and $|\mathcal{S}|$ are large. In this section, we further provide two techniques for enhancing CTS algorithms.

### 4.1    PARTIAL CACHING APPROXIMATION WITH TRANSFORMERS

In masked diffusion modeling, it is common to use a bidirectional transformer for computing logits at each position, where the standard key-value (KV) caching for transformers with causal masking (Katharopoulos et al., 2020) is not applicable. However, in CTS methods, we can use a similar idea to KV-caching for virtually increasing the sampling steps without proportionally increasing the cost.

Figure 2: Illustration of partial caching approximation applied to an $L$-layer transformer, where $\sigma = \mathrm{softmax}(\cdot/\sqrt{d_k})$, with $d_k$ being the dimension of key and query vectors.

Consider one CTS unmasking step. Let $[D]$ be the set of all the indices and $U \subset [D]$ be the set of already unmasked positions with values $\boldsymbol{x}_U$. To compute $p_{i|U}(\cdot|\boldsymbol{x}_U)$ for $i \in [D] \setminus U$, we input $D$ tokens to the transformer: $x_i$ at each position $i \in U$ and M at all the other positions. In ordinary CTS methods, we then sample $I \sim \pi(U, \boldsymbol{x}_U)$, the set of indices to unmask at this step, where $\pi$ can depend on the inferred $(p_{i|U}(\cdot|\boldsymbol{x}_U))_{i \in [D] \setminus U}$. Finally, we sample $x_i \sim p_{i|U}(\cdot|\boldsymbol{x}_U)$ for each $i \in I$.

Our *partial caching* method can incorporate arguably finer-grained token dependencies within $I$. First, we cache the key-value vectors as in the standard KV-caching. Then, divide $I$ into two as $A, B \subset I$ with $A \cup B = I$ and $A \cap B = \emptyset$. For $i \in A$, we just sample $x_i \sim p_{i|U}(\cdot|\boldsymbol{x}_U)$. We *approximate* sampling from $p_{i|U \cup A}(\cdot|\boldsymbol{x}_{U \cup A})$ for remaining indices $i \in B$ with low cost as follows:

- Run the transformer only at positions $i \in I$, with the input $x_i$ for $i \in A$ and M for $i \in B$.
- As key-value vectors at positions $i \notin I$, we use the cached ones.

It does not fully compute the effect of input change (from M to $x_i$ at each position $i \in A$) over positions other than $A \cup B$. However, it approximates the attention change for positions $i \in B$ that directly comes from $\boldsymbol{x}_A$, thus boosting the empirical performance (Figure 4). The second step operates with $|I|/D$ times computation of the original inference, and the empirical performance gain outweighs its additional computation. In practice, we can determine the partition $A \cup B = I$ by introducing an ordering. For instance, we let $A$ as the first half of $I$ in the order of confidence or preference. Figure 2 shows an example of such a two-stage sampling process. See Section 6 for comparison with other caching techniques for discrete diffusion.

## 4.2 BALANCING EXPLORATION AND EXPLOITATION

In this section, we decompose the error of a CTS algorithm and see there are two types of existing algorithms depending on which part of the error to optimize. We also propose a hybrid of the two.

Let us consider a two-round CTS algorithm. Suppose we have $N$ indices ($i \in [N]$) and will choose a size-$k$ subset $I$ in the first round. Let the resulting distribution of tokens over $\mathcal{S}^N$ be $p$ (ground truth: $q$), and we shall bound the KL divergence $D_{\mathrm{KL}}(q \,\|\, p)$. For each $I' \subset [N]$, let us define the distribution $\varphi(\cdot|I')$ over the subsets of $I'$ as $\varphi(J|I') = (|I'| + 1)^{-1}/\binom{|I'|}{|J|}$. Then, we have

$$D_{\mathrm{KL}}(q \,\|\, p) = D_{\mathrm{KL}}\left(q_I \,\middle\|\, \prod_{i \in I} q_i\right) + \mathbb{E}_{\boldsymbol{x}_I \sim q_I}\left[D_{\mathrm{KL}}\left(q_{I^c|I}(\cdot|\boldsymbol{x}_I) \,\middle\|\, \prod_{i \notin I} q_{i|I}(\cdot|\boldsymbol{x}_I)\right)\right]$$

$$\leq \underbrace{\sum_{i \in I} H(q_i)}_{\text{(a) exploitation}} - \underbrace{\sum_{i \in I} \mathbb{E}_{J \sim \varphi(\cdot|I \setminus \{i\}), \boldsymbol{x}_J \sim q_J}\left[H(q_{i|J}(\cdot|\boldsymbol{x}_J))\right]}_{\text{(b) spatial dispersion}} + \underbrace{\mathbb{E}_{\boldsymbol{x}_I \sim q_I}\left[\sum_{i \in [N] \setminus I} H(q_{i|I}(\cdot|\boldsymbol{x}_I))\right]}_{\text{(c) exploration}}, \quad (4)$$

where $H(\cdot)$ denotes the entropy of a probability distribution. The formal proof of (4) is given in Appendix C.2. After surveying existing approaches, we propose a "hybrid" approach balancing the optimization of different terms in (4).

Most of the existing adaptive unmasking schemes (Xiang et al., 2023; Zheng et al., 2024; Kim et al., 2025) just focus on greedy minimization of (4.a), if not explicitly based on entropy. Let us call this strategy *exploitation*. Term (4.b) is utilized by Ben-Hamu et al. (2025) for justifying adaptive selection of $k$, through its lower bound $\max_{i \in I} H(q_i)$, but it is not used for selecting $I$ itself, which is chosen through exploitation given $k$.

In contrast to exploitation methods, Halton-MaskGIT (Besnier et al., 2025) tries to maximize (4.b) by using the two-dimensional Halton sequence (Halton, 1960) for image modeling, which prevents the conditional entropy $\mathbb{E}_{\boldsymbol{x}_J \sim q_J}\big[H(q_{i|J}(\cdot|\boldsymbol{x}_J))\big]$ from becoming small (see Besnier et al., 2025, Section B). While not discussed, the use of low-discrepancy sequence including Halton sequence also aligns with the minimization of (4.c). Let us call *exploration* the strategy of using $I$ whose indices are non-informative to each other and informative as a set to the rest of indices.

**Proposed method: Hybrid approach.** As a natural consequence from the above observations, we propose a hybrid approach, taking balance between exploration and exploitation. Let $\boldsymbol{i} = (i_1, i_2, \ldots)$ and $\boldsymbol{j} = (j_1, j_2, \ldots)$ be two orderings of indices given by different strategies. To determine $n$ indices to unmask in the next step, we simply take the first $m$ $(< n)$ indices from $\boldsymbol{i}$ and follow the ordering of $\boldsymbol{j}$ for the rest to make a merged ordering $\boldsymbol{k}$. Here is an example in the case $n = 4$ and $m = 2$:

$$\boldsymbol{i} = (\underline{2}, \underline{3}, 6, 5, 1, 4), \quad \boldsymbol{j} = (\underline{4}, 3, \underline{1}, 5, 6, 2) \quad \Longrightarrow \quad \boldsymbol{k} = (\underline{2}, \underline{3}, \underline{4}, \underline{1}, 5, 6),$$

where underlined indices are chosen for the next unmasking step. In our experiment in Section 5.2, we use the Halton sequence (exploration) for $\boldsymbol{i}$ and moment-based ordering (exploitation) for $\boldsymbol{j}$. We can also apply the caching in the previous section by using the merged ordering.

## 5 EXPERIMENTS

We evaluate our theory and proposed methods on unconditional generation tasks in image and text domains. Additional experimental details are provided in Appendix D. The primary objectives of our experiments differ slightly between these two domains:

In Section 5.1, focusing on the image domain where MaskGIT already demonstrates strong performance, our aim is to validate our theory (that the moment sampler approximates MaskGIT; Theorem 2) and understand the role of temperature sampling. In the language domain (Section 5.2), however, we observe that temperature sampling significantly reduces generation diversity (Figure 5, Left). Therefore, we disable temperature sampling from CTS methods (i.e., $\gamma = 1$ in (CTS2)) for direct comparison of index selection algorithms. Under this setting, we further demonstrate that our hybridization (Section 4.2) improves sampler efficiency without compromising quality or diversity.

### 5.1 IMAGE MODELING

In our experiment in the image domain, we adopted MAGE (Li et al., 2023, ViT-B model) as the pretrained masked diffusion model. MAGE can be regarded as a masked diffusion model over a VQGAN (Esser et al., 2021) tokenizer space, trained on ImageNet $256 \times 256$ (Deng et al., 2009) for *un*conditional generation. Following the original implementation of MAGE, we employ a Gumbel temperature of $\alpha(1 - n/N)$ for the $n$-th step out of $N$ steps, where $\alpha$ is the global temperature parameter, and we adopt the cosine unmasking schedule (Chang et al., 2022). The schedules are shared within the tested methods. We compared **MaskGIT** (Algorithm 1) and **Moment** (with $\gamma = 1 + 1/\alpha$ in Algorithm 2, without temperature sampling in the final step) along with the following:

- **Temp**: We only conduct temperature sampling in **Moment**, i.e., replace the step (MM1) with uniformly random selection of indices.
- **Random**: Vanilla discrete diffusion sampler conditioned with fixed number of unmasking indices. It is equivalent to **MaskGIT** with $\alpha \to \infty$ or at-random index selection.
- **Halton**: Index ordering is given by a fixed two-dimensional Halton sequence as in (Besnier et al., 2025). There is no temperature involved in this sampler.

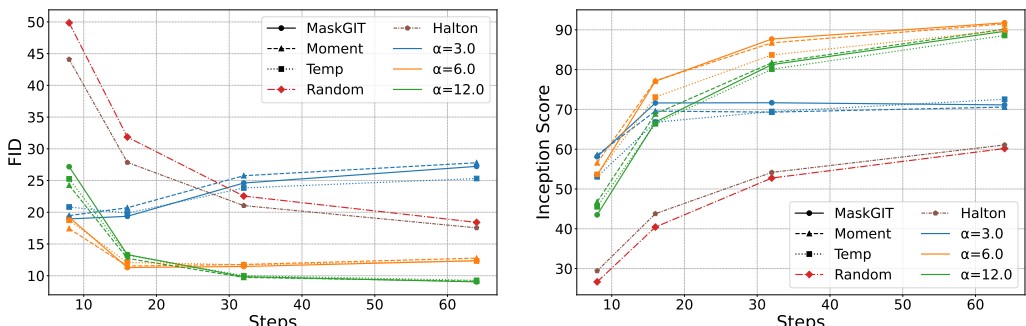

Figure 3: Fréchet Inception Distance (FID, ↓) and Inception Score (↑) against the number of steps for various samplers with MAGE. Both metrics were computed by 50,000 generated images. We can see that **Moment** closely approximates **MaskGIT** with the same temperature in both metrics; their quantitative approximation precision is given in Table A.

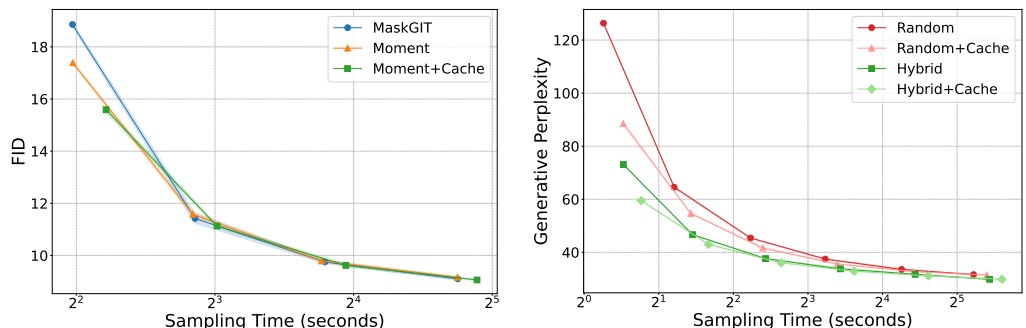

Figure 4: Average performance gains of our proposed samplers against sampling time per batch (on A6000 GPU). Each shaded region shows the standard deviation over three trials. (*Left*) FID of samplers applied to MAGE. (*Right*) Generative Perplexity (↓) of samplers applied to SDTT. Especially, **Hybrid** achieves approximately 2x speedup to achieve the same Generative Perplexity as **Random**.

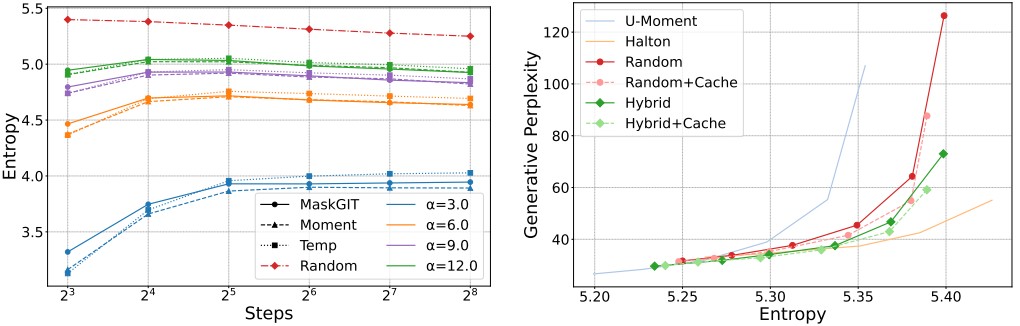

Figure 5: Language experiments. Each plot was computed by 1,024 generated sentences with 1,024 tokens with number of steps in $\{8, 16, 32, 64, 128, 256\}$. (*Left*) Entropy (↑) against the number of steps for temperature-based methods. We can see that temperature sampling significantly harms Entropy, as you can also see qualitatively in Appendix E.5.2. (*Right*) Trade-off between Generative Perplexity and Entropy. **Hybrid** samplers improve the tradeoff uniformly over **Random** across different numbers of steps.

Figure 3 shows the comparison of these five samplers combined with three choices of global Gumbel temperature: $\alpha \in \{3.0, 6.0, 12.0\}$ ($\alpha = 9.0$ is omitted to avoid making the figures too dense). Additionally, Table A shows the relative difference of **MaskGIT** and other samplers, in terms of the ImageNet FID performance. We can see that **Moment** shows similar performance to **MaskGIT**, supporting our theory. A surprising finding is that we can almost replicate the performance of **MaskGIT**

using `Temp` without adaptive ordering, which suggests that the performance of the MaskGIT sampler with MAGE primarily stems from its implicit temperature sampling rather than the confidence-based ordering. As a supplementary experiment, we compared these samplers in class-conditional setting, using MaskGIT-PyTorch (Besnier & Chen, 2023) as a pretrained masked model. The relative differences are reported in Table D in the appendix. In this setting, `Moment` again well approximates `MaskGIT` with the same temperature, while `Temp` is not necessarily aligned well with `MaskGIT`.

Table A: Mean relative difference (%) of ImageNet FIDs against reference MaskGIT samplers across 8, 16, 32, 64 steps in unconditional experiment with MAGE. Given reference sequence $(x_i)_i$, mean relative difference of sequence $(y_i)_i$ was computed as average of $|1 - y_i/x_i|$. Difference under 10% is bolded. `Moment` and `Temp` closely approximate `MaskGIT` with the same temperature.

| Ref. `MaskGIT` temperature | Random | MaskGIT | | | Moment | | | Temp | | |
|---|---|---|---|---|---|---|---|---|---|---|
| | | $\alpha = 3.0$ | 6.0 | 12.0 | 3.0 | 6.0 | 12.0 | 3.0 | 6.0 | 12.0 |
| $\alpha = 3.0$ | 67.1 | - | 37.8 | 50.4 | **4.1** | 38.5 | 47.3 | **5.7** | 36.4 | 47.5 |
| $\alpha = 6.0$ | 121.9 | 77.0 | - | 25.1 | 83.7 | **4.4** | 20.0 | 74.3 | **3.1** | 21.6 |
| $\alpha = 12.0$ | 113.9 | 106.8 | 24.5 | - | 113.4 | 27.6 | **4.3** | 98.8 | 24.0 | **3.0** |

In unconditional experiments with MAGE, we also tested our caching method in Figure 4(Left). `Moment+Cache` applies the caching technique in Section 4.1 to `Moment` for creating an "intermediate" step at each sampling step (see Appendix D.2 for details). For each sampler, we plotted the best FID among $\alpha \in \{3.0, 6.0, 9.0, 12.0\}$ for each number of sampling steps $N \in \{8, 16, 32, 64\}$, against the average sampling time per batch. The experimental result suggests that, while the performance gain might not be large enough to make the latency overhead negligible, the partial caching indeed gives some performance boost.

## 5.2 Language Modeling

In the language domain, we used SDTT (Deschenaux & Gulcehre, 2025, small model with KL divergence target) as the pretrained model, which was obtained after seven rounds of distillation of the MDLM model (Sahoo et al., 2024) trained on the OpenWebText dataset (Gokaslan & Cohen, 2019). SDTT is a masked diffusion model over the space of GPT-2 tokenizer (Radford et al., 2019). In the experiments, we adopted the linear unmasking schedule (i.e., unmasking the same number of positions at each step) and the same Gumbel temperature schedule as in Section 5.1. As evaluation metrics, we adopted Generative Perplexity (with GPT-2-large) and Entropy, which are in a trade-off relationship, following recent literature (Gat et al., 2024; Zheng et al., 2025).

We first compared `MaskGIT`, `Moment`, `Temp`, and `Random` as described in the previous section. We confirmed the tendency that the temperature roughly determines the performance; while the methods with temperature sampling attain lower (= better) Generative Perplexity (Figure 6, Left), their generated sentences have extremely lower Entropy (Figure 5, Left), thus harming the overall quality. To mitigate this issue, we introduced `U-Moment` (`U` for "unbiased"), where we only conduct index selection and omit the temperature sampling in `Moment`, i.e., the case of $\gamma = 1$ in Algorithm 2. In the experiment with "unbiased" methods (Figure 5, Right), `Halton` and `U-Moment` showed contrastive behaviors, where the former gave better trade-off with fewer steps, and the latter worked well with more steps. By merging these two into `Hybrid` as explained in Section 4.2, we obtain uniformly better trade-off compared to `Random` (see Appendix D.4.2 for the details of `Hybrid`).

By using `Hybrid` and/or `+Cache`, we can improve not only the trade-off (Figure 5, Right) but also sampling efficiency in terms of computational time (Figure 4, Right), leading to 1.5-2x acceleration. It shows the effectiveness of our approaches in Section 4. Note that, while the improvement of `Hybrid` seems consistent, the efficiency gain of `+Cache` can depend on GPUs (Appendix D.2.1).

## 6 Related Work

We overview relevant studies on efficient sampling/modeling for discrete diffusion.

**Efficient sampling schemes for discrete diffusion.** Several efficient sampling schemes have been proposed for speeding up discrete diffusion models. For general discrete diffusion models, not limited to masked diffusion, Park et al. (2025) proposed non-adaptive time-schedule optimization, and Ren et al. (2025) proposed discrete diffusion analogues of high-order ODE solvers. They can be combined with our method in principle.

For masked diffusion setting, Chang et al. (2022) proposed the MaskGIT sampler, which has been theoretically analyzed in this paper. LLaDA (Nie et al., 2025) utilizes the MaskGIT sampler in sampling from diffusion LLMs. Besnier et al. (2025) adopts the two-dimensional Halton sequence for selecting unmasking positions in masked image modeling (*exploration*). There have been also a few top-$k$ sampling (*exploitation*) methods including entropy (Xiang et al., 2023), confidence (Zheng et al., 2024), and probability margin (Kim et al., 2025). Additionally, Ben-Hamu et al. (2025) recently proposed an "entropy bounded" unmasking procedure that adaptively determines the number of unmasked indices in combination with any top-$k$ sampling criteria. See also Section 4.2 for how these methods can be understood from the viewpoints of exploration and exploitation. Finally, the first-hitting sampler (Zheng et al., 2025) demonstrated that the usual masked diffusion sampler can be accelerated by replacing it with an equivalent CTS sampler.

**Efficient modeling of dimensional correlation in discrete diffusion.** In contrast to the previous section, there are also methods that require additional training of models for efficient sampling in discrete diffusion. For general discrete diffusion, Hayakawa et al. (2025) proposed "mixture" modeling to distill dimensional correlations learned by many-step teacher models, and Chen et al. (2025) pointed out that quantum circuits can realize a one-step diffusion sampler.

For masked diffusion, Lezama et al. (2023) proposed the discrete predictor-corrector sampling, which is essentially based on learning an additional model that determines which token to unmask (see also Peng et al., 2025). Liu et al. (2025) and Xu et al. (2025) There are two methods exploiting a pretrained autoregressive model to recover dimensional correlations: one is based on copula (Liu et al., 2025), and the other conducts importance sampling (Xu et al., 2025), similar to speculative decoding in autoregressive models (Guo & Ermon, 2025). Finally, Zhu et al. (2025) distill pretrained masked image generation models into a one-step sampler.

**Caching methods for masked diffusion.** There are a couple of works on applying KV-cache to masked diffusion models such as Block Diffusion (Arriola et al., 2025) and dKV-Cache (Ma et al., 2025). They both focus on caching the KV information from past tokens for long-term efficiency, whereas our method in Section 4.1 caches past (already unmasked) and future (not unmasked in this round) tokens for stepwise efficiency.

## 7 CONCLUSION

In this paper, we have provided a theoretical analysis of the MaskGIT sampler for masked diffusion models, revealing its implicit temperature sampling mechanism. Through our analysis, we introduced the moment sampler, an asymptotically equivalent but more interpretable alternative that employs a "choose-then-sample" approach. We further enhanced the efficiency of general choose-then-sample algorithms through two techniques: a partial caching approximation for transformer-based models and a hybrid approach that formalizes the exploration-exploitation trade-off in adaptive unmasking. Our experiments in both image and language domains validated our theoretical findings and demonstrated the practical benefits of our proposed methods.

While our methods are post-hoc and can easily boost the efficiency of pretrained masked diffusion models whose outputs consist only of *one-token marginals*, more fundamental "modeling" approaches as in Section 6 are needed to accurately capture the complex distributions of high-dimensional discrete data. Theoretical understanding and practical improvement of such methods will be important future work toward realization of more efficient discrete generative models.

## LLM USAGE

We utilized LLMs for academic proofreading of the paper. We also used them for coding assistance, including algorithm implementation and result visualization.

ETHICS STATEMENT

While this work mainly presents theoretical contributions, the proposed methods could be applied to generate misleading or harmful content if used with inappropriately trained models. Furthermore, as our methods modify the sampling distribution, existing biases in pretrained models may be amplified or altered in unpredictable ways. We are committed to responsible use of these techniques and recommend implementing appropriate content filtering mechanisms in practical applications.

REPRODUCIBILITY STATEMENT

We provide the source code as supplementary material, and the detailed implementations of our experiments are described in Appendices A and D.

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

# A  ALGORITHMS

In this section, we itemize the algorithm pseudocodes of the MakGIT sampler (Algorithm 1), Moment Sampler (Algorithm 2), and the general form of choose-then-sample methods (Algorithm 3). To demonstrate the actual implementation of the moment sampler and highlight its differences from MaskGIT, we give their example PyTorch implementations in Figures A and B.

---

**Algorithm 1** One-round of MaskGIT sampler: $\text{OneRoundMaskGIT}((p_i)_{i \in I}, k, \alpha)$

---

**Require:**
    $(p_i)_{i \in I}$: Family of probability distributions over $\mathcal{S}$, with $|I| = N$
    $k \in [N]$: Number of indices to choose
    $\alpha > 0$: Gumbel temperature
**Ensure:**
    $(i_1, \ldots, i_k) \in I^k$: $k$ distinct indices
    $(x_{i_\ell})_{\ell=1}^k \in \mathcal{S}^k$: Sampled tokens
1: Independently sample standard Gumbel noise $\xi_i$ for each $i \in I$
2: Independently sample $x_i \sim p_i$ for each $i \in I$
3: $(i_1, \ldots, i_k) \leftarrow \text{argtop} k_{i \in I} \{\log p_i(x_i) + \alpha \xi_i\}$
4: **return** $(i_1, \ldots, i_k), (x_{i_\ell})_{\ell=1}^k$

---

**Algorithm 2** One-round of Moment sampler: $\text{OneRoundMoment}((p_i)_{i \in I}, k, \alpha, \gamma)$

---

**Require:**
    $(p_i)_{i \in I}$: Family of probability distributions over $\mathcal{S}$, with $|I| = N$
    $k \in [N]$: Number of indices to choose
    $\alpha > 0$: Gumbel temperature
    $\gamma > 0$: Inverse sampling temperature (set $\gamma = 1 + 1/\alpha$ when approximating MaskGIT)
**Ensure:**
    $(i_1, \ldots, i_k) \in I^k$: $k$ distinct indices
    $(x_{i_\ell})_{\ell=1}^k \in \mathcal{S}^k$: Sampled tokens
1: Independently sample standard Gumbel noise $\xi_i$ for each $i \in I$
2: $(i_1, \ldots, i_k) \leftarrow \text{argtop} k_{i \in I} \{\log\left(\sum_{x \in \mathcal{S}} p_i(x)^{1+1/\alpha}\right) + \xi_i\}$
3: Independently sample $x_{i_\ell} \sim p_{i_\ell}^\gamma / \|p_{i_\ell}\|_\gamma^\gamma$ for each $\ell \in [k]$
4: **return** $(i_1, \ldots, i_k), (x_{i_\ell})_{\ell=1}^k$

---

**Algorithm 3** General choose-then-sample algorithm

---

**Require:**
    $\pi(\cdot | I, \boldsymbol{x}_I)$: Distribution over nonempty subsets of $[D] \setminus I$ conditioned by $I$ and $\boldsymbol{x}_I$
    $p_{j|I}(\cdot | \boldsymbol{x}_I)$: Dimension-wise denoising model for any $j, I, \boldsymbol{x}_I$
    $\gamma > 0$: Inverse sampling temperature
**Ensure:**
    $\boldsymbol{x} \in \mathcal{S}^D$: Generated sample
1: Initialize $I \leftarrow \emptyset$
2: **while** $I \subsetneq [D]$ **do**
3:     Sample $J \sim \pi(\cdot | I, \boldsymbol{x}_I)$                             $\triangleright$ Ignore $\boldsymbol{x}_\emptyset = \emptyset$ when $I = \emptyset$
4:     Sample $x_j \sim \dfrac{p_{j|I}(\cdot | \boldsymbol{x}_I)^\gamma}{\|p_{j|I}(\cdot | \boldsymbol{x}_I)\|_\gamma^\gamma}$ for each $j \in J$     $\triangleright$ $\boldsymbol{x}_{I \cup J}$ has been determined so far
5:     $I \leftarrow I \cup J$
6: **end while**
7: **return** $\boldsymbol{x}$

---

```python
def maskgit_sampler(model, batch_size, seq_len, num_steps, mask_id, alpha):
    """
    MaskGIT sampler with multiple rounds

    Args:
        model: Prediction model
        batch_size: Batch size
        seq_len: Sequence length (D)
        num_steps: Number of sampling steps
        mask_id: Mask token ID in the vocabulary set
        alpha: Gumbel temperature for top-k selection
    """
    # Initialize: all tokens are masked
    x = torch.full((batch_size, seq_len), mask_id)
    # Compute k (assume num_steps is a divisor of seq_len)
    k = seq_len // num_steps

    for t in range(num_steps):
        # Get probability distribution p_i at each position
        logits = model(x)
        log_probs = torch.log_softmax(logits, dim=-1)  # log p_i

        # Step 1: Sample standard Gumbel noise for each i
        gumbel_noise = torch.distributions.Gumbel(0, 1).sample(
            (batch_size, seq_len)
        )

        # Step 2: Sample x_i from p_i for each i
        cat_dist = torch.distributions.Categorical(logits=logits)
        samples = cat_dist.sample()  # x_i's

        # Step 3: Select top-k indices (i_1, ..., i_k)
        # Get log p_i(x_i) for sampled tokens
        sampled_log_probs = torch.gather(
            log_probs, dim=-1, index=samples.unsqueeze(-1)
        ).squeeze(-1)  # log p_i(x_i)

        # Calculate confidence with Gumbel noise
        confidence = sampled_log_probs + alpha * gumbel_noise

        # Exclude already unmasked positions from top-k determination
        confidence[(x != mask_id)] = float('-inf')

        # Get top-k indices with highest confidence (to unmask)
        _, indices_to_unmask = torch.topk(
            confidence, k, largest=True
        )

        # Create boolean mask for indices to unmask
        unmask = torch.zeros_like(x, dtype=torch.bool)
        unmask.scatter_(1, indices_to_unmask, True)

        # Step 4: Update only the selected positions
        x[unmask] = samples[unmask]

    return x
```

Figure A: Example PyTorch implementation of MaskGIT sampler. For simplicity, we unmask the same number of positions at each step, and the Gumbel temperature is held constant throughout sampling iterations. The "Steps" in the comments correspond to those in Algorithm 1, and the highlighted lines show the differences from Moment sampler in Figure B.

```python
def moment_sampler(model, batch_size, seq_len,
                   num_steps, mask_id, alpha, gamma):
    """
    Moment sampler with multiple rounds

    Args:
        model, ..., alpha: Same as maskgit_sampler
                    (alpha is assumed positive for simplicity)
        gamma: Inverse sampling temperature
                (set gamma = 1 + 1/alpha when approximating MaskGIT)
    """
    # Initialize: all tokens are masked
    x = torch.full((batch_size, seq_len), mask_id)
    # Compute k (assume num_steps is a divisor of seq_len)
    k = seq_len // num_steps

    for t in range(num_steps):
        # Get probability distribution p_i at each position
        logits = model(x)
        log_probs = torch.log_softmax(logits, dim=-1)  # log p_i

        # Step 1: Sample standard Gumbel noise for each i
        gumbel_noise = torch.distributions.Gumbel(0, 1).sample(
            (batch_size, seq_len)
        )

        # Step 2: Select top-k indices (i_1, ..., i_k)
        # Compute log(sum_x p_i(x)^(1+1/alpha)) for each i
        moment_scores = torch.logsumexp(
            (1 + 1/alpha) * log_probs, dim=-1
        )

        # Calculate confidence with Gumbel noise
        confidence = moment_scores + gumbel_noise

        # Exclude already unmasked positions from top-k determination
        confidence[(x != mask_id)] = float('-inf')

        # Get top-k indices with highest confidence (to unmask)
        _, indices_to_unmask = torch.topk(
            confidence, k, largest=True
        )

        # Step 3: Sample x_{i_ell} from p_{i_ell}^gamma for each ell
        # Extract log-probs only at positions to unmask ([batch_size, k])
        batch_indices = torch.arange(batch_size).unsqueeze(1).expand(-1, k)
        selected_log_probs = log_probs[batch_indices, indices_to_unmask]

        # Sample from the distribution proportional to p_{i_ell}^gamma
        cat_dist = torch.distributions.Categorical(
            logits=gamma * selected_log_probs
        )
        samples = cat_dist.sample()

        # Step 4: Update only the selected positions
        x[batch_indices, indices_to_unmask] = samples

    return x
```

Figure B: Example PyTorch implementation of moment sampler. For simplicity, we unmask the same number of positions at each step, and the Gumbel temperature is held constant throughout sampling iterations. The "Steps" in the comments correspond to those in Algorithm 2, and the highlighted lines show the differences from MaskGIT sampler in Figure A.

## B   FORMAL AND INFORMAL DERIVATION OF MOMENT SAMPLER

For notational simplicity in the proofs, let $\beta := 1 + 1/\alpha$ throughout this section.

### B.1   INFORMAL DERIVATION OF MOMENT SAMPLER

In this section, we derive the moment sampler approximation (3) from (1) and (2). First, by applying the approximation (2) to (1), we have

$$\mathbb{P}\big(i_t^* = i_t \mid i_1^* = i_1, \ldots, i_{t-1}^* = i_{t-1}, (x_i)_{i=1}^N\big) \approx \frac{p_{i_t}(x_{i_t})^{1/\alpha}}{\sum_{i \in [N] \setminus I_{t-1}} \|p_i\|_\beta^\beta}, \tag{5}$$

for each $t = 1, \ldots, k$. Then, by multiplying this for $t = 1, \ldots, \ell - 1$, we obtain

$$\mathbb{P}\big(i_1^* = i_1, \ldots, i_{\ell-1}^* = i_{\ell-1} \mid (x_i)_{i=1}^N\big) \approx \prod_{t=1}^{\ell-1} \frac{p_{i_t}(x_{i_t})^{1/\alpha}}{\sum_{i \in [N] \setminus I_{t-1}} \|p_i\|_\beta^\beta}.$$

Note that the right-hand side is independent of $(x_i)_{i \in [N] \setminus I_\ell}$, so we can replace the conditioning on $(x_i)_{i=1}^N$ by $(x_i)_{i \in I_\ell}$, By marginalizing out $x_i$ with $i \in I_{\ell-1}$ from this, we have

$$\mathbb{P}(i_1^* = i_1, \ldots, i_\ell^* = i_\ell \mid x_{i_\ell}) = \sum_{x_{i_1}} \cdots \sum_{x_{i_{\ell-1}}} \mathbb{P}(i_1^* = i_1, \ldots, i_\ell^* = i_\ell \mid (x_i)_{i \in I_\ell}) \mathbb{P}\big((x_i)_{i \in I_{\ell-1}} \mid x_{i_\ell}\big)$$

$$\approx \left( \prod_{t=1}^{\ell-1} \frac{\sum_{x_{i_t}} p_{i_t}(x_{i_t})^{1/\alpha} \cdot p_{i_t}(x_{i_t})}{\sum_{i \in [N] \setminus I_{t-1}} \|p_i\|_\beta^\beta} \right) \frac{p_{i_\ell}(x_{i_\ell})^{1/\alpha}}{\sum_{i \in [N] \setminus I_{\ell-1}} \|p_i\|_\beta^\beta}$$

$$= \underbrace{\left( \prod_{t=1}^{\ell-1} \frac{\|p_{i_t}\|_\beta^\beta}{\sum_{i \in [N] \setminus I_{t-1}} \|p_i\|_\beta^\beta} \right)}_{=: C_{\ell-1}(i_1, \ldots, i_{\ell-1})} \frac{p_{i_\ell}(x_{i_\ell})^{1/\alpha}}{\sum_{i \in [N] \setminus I_{\ell-1}} \|p_i\|_\beta^\beta}.$$

Therefore, we have

$$\mathbb{P}\big(x_{i_\ell}, i_\ell^* = i_\ell \mid i_1^* = i_1, \ldots, i_{\ell-1}^* = i_{\ell-1}\big) = \frac{\mathbb{P}(i_1^* = i_1, \ldots, i_\ell^* = i_\ell \mid x_{i_\ell}) \mathbb{P}(x_{i_\ell})}{\mathbb{P}\big(i_1^* = i_1, \ldots, i_{\ell-1}^* = i_{\ell-1}\big)}$$

$$\approx \frac{C_{\ell-1}(i_1, \ldots, i_{\ell-1})}{\mathbb{P}(i_1^* = i_1, \ldots, i_{\ell-1}^* = i_{\ell-1})} \frac{p_{i_\ell}(x_{i_\ell})^\beta}{\sum_{i \in [N] \setminus I_{\ell-1}} \|p_i\|_\beta^\beta} \propto \frac{p_{i_\ell}(x_{i_\ell})^\beta}{\sum_{i \in [N] \setminus I_{\ell-1}} \|p_i\|_\beta^\beta}.$$

Since the final right-hand side is actually a normalized probability distribution over $\mathcal{S} \times ([N] \setminus I_{\ell-1})$, we have derived (3).

### B.2   STRATEGY FOR FORMAL PROOF OF THEOREM 2

Formally, we work under the following setting:

**Setting A** (MaskGIT sampler). *We are given $\alpha > 0$ and $N$ probability distributions $p_1, \ldots, p_N$ over a finite set $\mathcal{S}$. We sample $x_i \sim p_i$ and standard Gumbel noise $\xi_i$ independently for each $i \in [N]$, and let $(i_1^*, \ldots, i_k^*) = \operatorname{argtop} k_{i \in [N]} \{\log p_i(x_i) + \alpha \xi_i\}$ with $k \in [N]$.*

Let us start by applying Bernstein's inequality (see, e.g., Boucheron et al. 2013, Corollary 2.11) to the sum of $p_i(x_i)^{1/\alpha} - \mathbb{E}\big[p_i(x_i)^{1/\alpha}\big]$. Since the summand is always within $[-1, 1]$, for $t \geq 0$, we have

$$\mathbb{P}\left( \sum_{i=1}^N p_i(x_i)^{1/\alpha} - \sum_{i=1}^N \mathbb{E}\big[p_i(x_i)^{1/\alpha}\big] \leq -t \right) \leq \exp\left( -\frac{t^2}{2(\sigma^2 + t/3)} \right), \tag{6}$$

where $\sigma^2 := \sum_{i=1}^N \mathbb{E}\big[\big(p_i(x_i)^{1/\alpha} - \mathbb{E}\big[p_i(x_i)^{1/\alpha}\big]\big)^2\big]$. Since $0 \leq p_i(x_i)^{1/\alpha} \leq 1$, we have

$$\sigma^2 \leq \sum_{i=1}^N \mathbb{E}\big[p_i(x_i)^{1/\alpha} \cdot p_i(x_i)^{1/\alpha}\big] \leq \sum_{i=1}^N \mathbb{E}\big[p_i(x_i)^{1/\alpha}\big]. \tag{7}$$

Now, recall that $\mathbb{E}\left[p_i(x_i)^{1/\alpha}\right] = \|p_i\|_\beta^\beta$ for $\beta = 1 + 1/\alpha$. By letting $t = \epsilon \sum_{i=1}^N \|p_i\|_\beta^\beta$ for some $0 \le \epsilon \le 1$ and combining (6) and (7), we have

$$
\mathbb{P}\left(\sum_{i=1}^N p_i(x_i)^{1/\alpha} \le (1-\epsilon) \sum_{i=1}^N \|p_i\|_\beta^\beta\right) \le \exp\left(-\frac{\epsilon^2 \left(\sum_{i=1}^N \|p_i\|_\beta^\beta\right)^2}{2(\sum_{i=1}^N \|p_i\|_\beta^\beta + \epsilon \sum_{i=1}^N \|p_i\|_\beta^\beta/3)}\right)
$$

$$
= \exp\left(-\frac{\epsilon^2 \left(\sum_{i=1}^N \|p_i\|_\beta^\beta\right)}{2(1+\epsilon/3)}\right)
$$

$$
\le \exp\left(-\frac{3\epsilon^2}{8} \sum_{i=1}^N \|p_i\|_\beta^\beta\right). \tag{8}
$$

To utilize this estimate for our analysis of MaskGIT sampler, we prove the following proposition. The proof is given in Appendix C.3.

**Proposition 4.** *Under Setting A, let $[N]_{<k}$ denote the set of all the subsets of $[N]$ with cardinality less than $k$. Then, for $\epsilon \in [0, 1]$, we have*

$$
\sum_{i \in [N] \setminus I} p_i(x_i)^{1/\alpha} > \left(1 - \epsilon - \frac{(k-1)S^{1/\alpha}}{N-k+1}\right) \sum_{i \in [N] \setminus I} \|p_i\|_\beta^\beta \quad \text{for all } I \in [N]_{<k} \tag{9}
$$

*with probability at least $1 - \exp\left(-\frac{3}{8}\epsilon^2 N |\mathcal{S}|^{-1/\alpha}\right)$.*

We use Proposition 4 to prove the following assertion, whose proof is given in Appendix C.4

**Proposition 5.** *Under Setting A, assume $\epsilon + \frac{(k-1)|\mathcal{S}|^{1/\alpha}}{N-k+1} < 1$ holds. For $\epsilon \in [0, 1]$, let us define $\mathcal{Z}_\epsilon \subset \mathcal{S}^N$ as the set of $(z_i)_{i=1}^N \in \mathcal{S}^N$ that satisfies the following inequality:*

$$
\mathbb{P}\left(i_1^* = i_1, \ldots, i_k^* = i_k \mid (x_i)_{i=1}^N = (z_i)_{i=1}^N\right)
$$

$$
< \left(1 - \epsilon - \frac{(k-1)|\mathcal{S}|^{1/\alpha}}{N-k+1}\right)^{-k} \prod_{\ell=1}^k \frac{p_{i_\ell}(z_{i_\ell})^{1/\alpha}}{\sum_{i \in [N] \setminus I_{\ell-1}} \|p_i\|_\beta^\beta},
$$

*where $I_{\ell-1} := \{i_1, \ldots, i_{\ell-1}\}$. Then, we have $\mathbb{P}\left((x_i)_{i=1}^N \in \mathcal{Z}_\epsilon\right) \ge 1 - \exp\left(-\frac{3}{8}\epsilon^2 N |\mathcal{S}|^{-1/\alpha}\right)$.*

We next compute the probability distribution of the moment sampler.

**Setting B** (Moment sampler). *We are given $\alpha > 0$ and $N$ probability distributions $p_1, \ldots, p_N$ over a finite set $\mathcal{S}$. Let $\beta = 1 + 1/\alpha$. We sample $y_i \sim p_i^\beta / \|p_i\|_\beta^\beta$ and standard Gumbel noise $\eta_i$ independently for each $i \in [N]$, and let $(j_1^*, \ldots, j_k^*) = \operatorname{argtop} k_{i \in [N]}\{\log \|p_i\|_\beta^\beta + \eta_i\}$ with $k \in [N]$.*

**Proposition 6.** *Under Setting B, for each distinct indices $i_1, \ldots, i_k \in [N]$ and (not necessarily distinct) $z_{i_1}, \ldots, z_{i_k} \in \mathcal{S}$, we have*

$$
\mathbb{P}\left(j_1^* = i_1, \ldots, j_k^* = i_k, (y_{j_\ell^*})_{\ell=1}^k = (z_{i_\ell})_{\ell=1}^k\right) = \prod_{\ell=1}^k \frac{p_{i_\ell}(z_{i_\ell})^\beta}{\sum_{i \in [N] \setminus I_{\ell-1}} \|p_i\|_\beta^\beta} \tag{10}
$$

*where $I_{\ell-1} := \{i_1, \ldots, i_{\ell-1}\}$.*

The proof is given in Appendix C.5.

Let us finally prove Theorem 2. The following is its restatement under the above settings.

**Theorem 7.** *Under Setting A & B, for each $\boldsymbol{i} = (i_1, \ldots, i_k) \in [N]^k$ and $(z_{i_\ell})_{\ell=1}^k \in \mathcal{S}^k$, let*

$$
p_{\text{MaskGIT}}(\boldsymbol{i}, (z_{i_\ell})_{\ell=1}^k) := \mathbb{P}\left(i_1^* = i_1, \ldots, i_k^* = i_k, (x_{i_\ell^*})_{\ell=1}^k = (z_{i_\ell})_{\ell=1}^k\right),
$$

$$
p_{\text{moment}}(\boldsymbol{i}, (z_{i_\ell})_{\ell=1}^k) := \mathbb{P}\left(j_1^* = i_1, \ldots, j_k^* = i_k, (y_{j_\ell^*})_{\ell=1}^k = (z_{i_\ell})_{\ell=1}^k\right).
$$

Then, as probability distributions over $[N]^k \times \mathcal{S}^k$, we have

$$d_{\mathrm{TV}}(p_{\mathrm{moment}}, p_{\mathrm{MaskGIT}}) \leq 5\sqrt{\frac{k^2 |\mathcal{S}|^{1/\alpha}}{N}} \left(1 + \sqrt{\log^+\left(\frac{N}{k^2 |\mathcal{S}|^{1/\alpha}}\right)}\right),$$

where $\log^+(x) := \log(\max\{1, x\})$ for $x \in \mathbb{R}$.

This main result is proven in Appendix C.6.

## C  PROOFS

### C.1  PROOF OF PROPOSITION 3

*Proof.* Let $I_0 = \emptyset$ and inductively (and randomly) define

$$\sigma_n \sim \pi(\cdot | \boldsymbol{x}_{I_{n-1}}), \qquad x_{\sigma_n} \sim p_{\sigma_n | I_{n-1}}(\cdot | \boldsymbol{x}_{I_{n-1}}), \qquad I_n := I_{n-1} \cup \{\sigma_n\}$$

for $n = 1, \ldots, D$, where we abuse the notation to simplify $\{\sigma_n\} \sim \pi$ into $\sigma_n \sim \pi$. Note that $(\sigma_1, \ldots, \sigma_D)$ is a random permutation of $(1, \ldots, D)$. Let us write $p(\boldsymbol{x}, \boldsymbol{\sigma})$ be the joint distribution of $\boldsymbol{x}$ and $\boldsymbol{\sigma} = (\sigma_n)_{n=1}^D$ (the latter is constrained to be a permutation). Then, we have

$$p(\boldsymbol{x}, \boldsymbol{\sigma}) = \prod_{n=1}^D \pi(\sigma_n | \boldsymbol{x}_{I_{n-1}}) p_{\sigma_n | I_{n-1}}(x_{\sigma_n} | \boldsymbol{x}_{I_{n-1}})$$

$$= \left(\prod_{n=1}^D \pi(\sigma_n | \boldsymbol{x}_{I_{n-1}})\right) \left(\prod_{n=1}^D p_{\sigma_n | I_{n-1}}(x_{\sigma_n} | \boldsymbol{x}_{I_{n-1}})\right) = q_{\mathrm{data}}(\boldsymbol{x}) \prod_{n=1}^D \pi(\sigma_n | \boldsymbol{x}_{I_{n-1}})$$

from the assumption $p_{j|I} = q_{j|I}$. Thus, it suffices to prove that $\sum_{\boldsymbol{\sigma}} \prod_{n=1}^D \pi(\sigma_n | \boldsymbol{x}_{I_{n-1}}) = 1$. To this end, let us prove the following for $k = 1, \ldots, D$ by induction on $k$ (the case $k = D$ is what we would like to prove ultimately):

$$\sum_{\sigma_{D-k+1}, \ldots, \sigma_D} \prod_{n=D-k+1}^D \pi(\sigma_n | \boldsymbol{x}_{I_{n-1}}) = 1 \qquad \text{for any } I_{D-k} \text{ with } |I_{D-k}| = D - k \text{ and } \boldsymbol{x}_{I_{D-k}}.$$

$$\tag{11}$$

It is clearly true for $k = 1$, since it is just the total probability sum of the probability distribution $\pi(\cdot | \boldsymbol{x}_{I_{D-1}})$. For $k \geq 2$, from the induction hypothesis, we have

$$\sum_{\sigma_{D-k+1}, \ldots, \sigma_D} \prod_{n=D-k+1}^D \pi(\sigma_n | \boldsymbol{x}_{I_{n-1}}) = \sum_{\sigma_{D-k+1}} \pi(\sigma_{D-k+1} | \boldsymbol{x}_{I_{D-k}}) \underbrace{\sum_{\sigma_{D-k+2}, \ldots, \sigma_D} \prod_{n=D-k+1}^D \pi(\sigma_n | \boldsymbol{x}_{I_{n-1}})}_{=1 \text{ by induction hypothesis}}$$

$$= \sum_{\sigma_{D-k+1}} \pi(\sigma_{D-k+1} | \boldsymbol{x}_{I_{D-k}}) = 1.$$

Therefore, the proof has been completed. $\qquad\square$

### C.2  PROOF OF EQUATION 4

*Proof.* By using the chain rule of KL divergence (Cover & Thomas, 2006, Theorem 2.5.3), we have

$$D_{\mathrm{KL}}(q \| p) = D_{\mathrm{KL}}(q_I \| p_I) + \mathbb{E}_{\boldsymbol{x}_I \sim q_I}\left[D_{\mathrm{KL}}(q_{I^c | I}(\cdot | \boldsymbol{x}_I) \| p_{I^c | I}(\cdot | \boldsymbol{x}_I))\right], \tag{12}$$

which shows the first inequality in (4). Let us first consider the KL divergence between $q_I$ and $p_I$. First, we have

$$D_{\mathrm{KL}}(q_I \| p_I) = \mathbb{E}_{\boldsymbol{x}_I \sim q_I}\left[-\log\left(\prod_{i \in I} q_i(x_i)\right) + \log q_I(\boldsymbol{x}_I)\right]$$

$$= \sum_{i \in I} \mathbb{E}_{x_i \sim q_i}[-\log q_i(x_i)] - \mathbb{E}_{\boldsymbol{x}_I \sim q_I}[-\log q_I(\boldsymbol{x}_I)]$$

$$= \sum_{i \in I} H(q_i) - \mathbb{E}_{\boldsymbol{x}_I \sim q_I}[-\log q_I(\boldsymbol{x}_I)]. \tag{13}$$

Next, by using a permutation $(\sigma_1, \ldots, \sigma_k)$ of $I$, the remainder term of (13) (entropy of $q_I$) can be rewritten as follows:

$$
\mathbb{E}_{\boldsymbol{x}_I \sim q_I}[-\log q_I(\boldsymbol{x}_I)] = \sum_{x_{\sigma_1}, \ldots, x_{\sigma_k}} \prod_{j=1}^k q_{\sigma_j | I_{j-1}}(x_{\sigma_j} | \boldsymbol{x}_{I_{j-1}}) \sum_{\ell=1}^k (-\log q_{\sigma_\ell | I_{\ell-1}}(x_{\sigma_\ell} | \boldsymbol{x}_{I_{\ell-1}}))
$$

$$
= \sum_{\ell=1}^k \sum_{x_{\sigma_1}, \ldots, x_{\sigma_\ell}} q_{I_{\ell-1}}(\boldsymbol{x}_{I_{\ell-1}}) q_{\sigma_\ell | I_{\ell-1}}(x_{\sigma_\ell} | \boldsymbol{x}_{I_{\ell-1}})(-\log q_{\sigma_\ell | I_{\ell-1}}(x_{\sigma_\ell} | \boldsymbol{x}_{I_{\ell-1}}))
$$

$$
= \sum_{\ell=1}^k \mathbb{E}_{\boldsymbol{x}_{I_{\ell-1}} \sim q_{I_{\ell-1}}} \mathbb{E}_{x_{\sigma_\ell} \sim q_{\sigma_\ell | I_{\ell-1}}} \left[ -\log q_{\sigma_\ell | I_{\ell-1}}(x_{\sigma_\ell} | \boldsymbol{x}_{I_{\ell-1}}) \right]
$$

$$
= \sum_{\ell=1}^k \mathbb{E}_{\boldsymbol{x}_{I_{\ell-1}} \sim q_{I_{\ell-1}}} \left[ H(q_{\sigma_\ell | I_{\ell-1}}(\cdot | \boldsymbol{x}_{I_{\ell-1}})) \right]. \tag{14}
$$

Let us consider taking the average of the right-hand side over all the permutations. With the uniformly random permutation $(\sigma_1, \ldots, \sigma_k)$, for each $i \in I$, $\ell$ with $\sigma_\ell = i$ takes the uniformly distribution over $[k]$. $I_{\ell-1}$ (conditioned by $\ell$) then takes the uniform distribution over all the possible size-$(\ell-1)$ subsets of $I \setminus \{i\}$. Therefore, if we write such an $\ell$ as $\ell = \sigma^{-1}(i)$, the probability that $J \subset I \setminus \{i\}$ is chosen as $I_{\sigma^{-1}(i)-1}$ can be computed as

$$
\underbrace{\frac{1}{k}}_{\mathbb{P}(\sigma^{-1}(i)-1=|J|)} \binom{k-1}{|J|}^{-1} = \varphi(J | I \setminus \{i\}),
$$

where $\varphi$ is the probability distribution defined just before (4). By applying this to (14), we have

$$
\mathbb{E}_{\boldsymbol{x}_I \sim q_I}[-\log q_I(\boldsymbol{x}_I)] = \sum_{i \in I} \mathbb{E}_{J \sim \varphi(\cdot | I \setminus \{i\})} \mathbb{E}_{\boldsymbol{x}_J \sim q_J} \left[ H(q_{i|J}(\cdot | \boldsymbol{x}_J)) \right].
$$

Combining it with (13), we obtain

$$
D_{\mathrm{KL}}(q_I \,\|\, p_I) = \sum_{i \in I} H(q_i) - \sum_{i \in I} \mathbb{E}_{J \sim \varphi(\cdot | I \setminus \{i\})} \mathbb{E}_{\boldsymbol{x}_J \sim q_J} \left[ H(q_{i|J}(\cdot | \boldsymbol{x}_J)) \right]. \tag{15}
$$

Finally, for the remaining term, it suffices to prove

$$
\mathbb{E}_{\boldsymbol{x}_I \sim q_I} \left[ D_{\mathrm{KL}}(q_{I^c|I}(\cdot | \boldsymbol{x}_I) \,\|\, p_{I^c|I}(\cdot | \boldsymbol{x}_I)) \right] \leq \mathbb{E}_{\boldsymbol{x}_I \sim q_I} \left[ \sum_{i \in [N] \setminus I} H(q_{i|I}(\cdot | \boldsymbol{x}_I)) \right].
$$

This can be proven by using the positivity of entropy and modifying (15), with $q$ replaced by $q_{\cdot|I}$ and $I$ replaced by $I^c$. Thus, the proof is completed. $\qquad \square$

### C.3 PROOF OF PROPOSITION 4

*Proof.* Let $\Omega_\epsilon$ be the event under which we have $\sum_{i=1}^N p_i(x_i)^{1/\alpha} > (1-\epsilon) \sum_{i=1}^N \|p_i\|_\beta^\beta$. From (8), we have $\mathbb{P}(\Omega_\epsilon) \geq 1 - \exp\left(-\frac{3\epsilon^2}{8} \sum_{i=1}^N \|p_i\|_\beta^\beta\right)$. For any $I \in [N]_{<k}$, under $\Omega_\epsilon$, we have

$$
\sum_{i \in [N] \setminus I} p_i(x_i)^{1/\alpha} = \sum_{i=1}^N p_i(x_i)^{1/\alpha} - \sum_{i \in I} p_i(x_i)^{1/\alpha}
$$

$$
> (1-\epsilon) \sum_{i=1}^N \|p_i\|_\beta^\beta - \sum_{i \in I} p_i(x_i)^{1/\alpha}
$$

$$
\geq (1-\epsilon) \sum_{i \in [N] \setminus I} \|p_i\|_\beta^\beta - \sum_{i \in I} p_i(x_i)^{1/\alpha} \tag{16}
$$

Now we want to estimate the ratio between $\sum_{i \in I} p_i(x_i)^{1/\alpha}$ and $\sum_{i \in [N] \setminus I} \|p_i\|_\beta^\beta$. Let $S := |\mathcal{S}|$. From Hölder's inequality, we have

$$\|p_i\|_\beta^\beta = S \cdot \frac{1}{S} \sum_{x \in \mathcal{S}} p_i(x)^\beta \geq S \left( \frac{1}{S} \sum_{x \in \mathcal{S}} p_i(x) \right)^\beta = S^{1-\beta} = S^{-1/\alpha}. \tag{17}$$

By using this, we obtain

$$\frac{\sum_{i \in I} p_i(x_i)^{1/\alpha}}{\sum_{i \in [N] \setminus I} \|p_i\|_\beta^\beta} \leq \frac{(k-1)}{(N-k+1)S^{-1/\alpha}} = \frac{(k-1)S^{1/\alpha}}{N-k+1}.$$

By applying this to (16), under $\Omega_\epsilon$, we have

$$\sum_{i \in [N] \setminus I} p_i(x_i)^{1/\alpha} > \left( 1 - \epsilon - \frac{(k-1)S^{1/\alpha}}{N-k+1} \right) \sum_{i \in [N] \setminus I} \|p_i\|_\beta^\beta.$$

Moreover, by using (17), we have

$$\mathbb{P}(\Omega_\epsilon) \geq 1 - \exp\left( -\frac{3\epsilon^2}{8} \sum_{i=1}^N \|p_i\|_\beta^\beta \right) \geq 1 - \exp\left( -\frac{3\epsilon^2}{8} N S^{-1/\alpha} \right),$$

which completes the proof of the proposition. $\qquad\square$

### C.4 PROOF OF PROPOSITION 5

*Proof.* Whether or not (9) holds only depends on the actual values of $(x_i)_{i=1}^N$. Thus, we can define the set $\mathcal{Z}'_\epsilon \subset \mathcal{S}^N$ such that (9) is satisfied if and only if $(x_i)_{i=1}^N \in \mathcal{Z}'_\epsilon$. Now, let $(z_i)_{i=1}^N \in \mathcal{Z}'_\epsilon$. From (1) and (9), we have (recall $S = |\mathcal{S}|$)

$$\mathbb{P}\big( i_1^* = i_1, \ldots, i_k^* = i_k \mid (x_i)_{i=1}^N = (z_i)_{i=1}^N \big)$$

$$= \prod_{\ell=1}^k \mathbb{P}\big( i_\ell^* = i_\ell \mid i_1^* = i_1, \ldots, i_{\ell-1}^* = i_{\ell-1}, (x_i)_{i=1}^N = (z_i)_{i=1}^N \big)$$

$$= \prod_{\ell=1}^k \frac{p_{i_\ell}(x_{i_\ell})^{1/\alpha}}{\sum_{i \in [N] \setminus I_{\ell-1}} p_i(x_i)^{1/\alpha}}$$

$$< \prod_{\ell=1}^k \frac{p_{i_\ell}(x_{i_\ell})^{1/\alpha}}{\left( 1 - \epsilon - \frac{(k-1)S^{1/\alpha}}{N-k+1} \right) \sum_{i \in [N] \setminus I_{\ell-1}} \|p_i\|_\beta^\beta}$$

$$= \left( 1 - \epsilon - \frac{(k-1)S^{1/\alpha}}{N-k+1} \right)^{-k} \prod_{\ell=1}^k \frac{p_{i_\ell}(x_{i_\ell})^{1/\alpha}}{\sum_{i \in [N] \setminus I_{\ell-1}} \|p_i\|_\beta^\beta}.$$

We thus have $z \in \mathcal{Z}_\epsilon$, which implies $\mathcal{Z}'_\epsilon \subset \mathcal{Z}_\epsilon$. Therefore,

$$\mathbb{P}\big( (x_i)_{i=1}^N \in \mathcal{Z}_\epsilon \big) \geq \mathbb{P}\big( (x_i)_{i=1}^N \in \mathcal{Z}'_\epsilon \big) \geq 1 - \exp\left( -\frac{3}{8} \epsilon^2 N |\mathcal{S}|^{-1/\alpha} \right)$$

follows from Proposition 4 and the definition of $\mathcal{Z}'_\epsilon$. $\qquad\square$

### C.5 PROOF OF PROPOSITION 6

*Proof.* From the independence of $y_i$ and $(j_\ell^*)_{\ell=1}^k$ in Setting B, we have

$$\mathbb{P}\big( j_1^* = i_1, \ldots, j_k^* = i_k, (y_{j_\ell^*})_{\ell=1}^k = (z_{i_\ell})_{\ell=1}^k \big)$$

$$= \mathbb{P}\big( j_1^* = i_1, \ldots, j_k^* = i_k \big) \, \mathbb{P}\big( (y_{i_\ell})_{\ell=1}^k = (z_{i_\ell})_{\ell=1}^k \mid j_1^* = i_1, \ldots, j_k^* = i_k \big)$$

$$= \mathbb{P}\big( j_1^* = i_1, \ldots, j_k^* = i_k \big) \prod_{\ell=1}^k \mathbb{P}(y_{i_\ell} = z_{i_\ell})$$

$$= \mathbb{P}\big( j_1^* = i_1, \ldots, j_k^* = i_k \big) \prod_{\ell=1}^k \frac{p_{i_\ell}(z_{i_\ell})^\beta}{\|p_{i_\ell}\|_\beta^\beta}.$$

Now, we have

$$\mathbb{P}(j_1^* = i_1, \dots, j_k^* = i_k) = \prod_{\ell=1}^{k} \frac{\|p_{i_\ell}\|_\beta^\beta}{\sum_{i \in [N] \setminus I_{\ell-1}} \|p_i\|_\beta^\beta}$$

by letting $\mu_i = \log\|\mu_i\|_\beta^\beta$ in Proposition 1. We obtain (10) through these two identities. □

### C.6 Proof of Theorem 7

*Proof.* Let $\boldsymbol{z} \in \mathcal{S}^N$. By introducing extra variables, let us define

$$\tilde{p}_{\text{MaskGIT}}(\boldsymbol{i}, \boldsymbol{z}) := \mathbb{P}\big(i_1^* = i_1, \dots, i_k^* = i_k, (x_i)_{i=1}^N = (z_i)_{i=1}^N\big),$$
$$\tilde{p}_{\text{moment}}(\boldsymbol{i}, \boldsymbol{z}) := \mathbb{P}\big(j_1^* = i_1, \dots, j_k^* = i_k, (y_{i_\ell})_{\ell=1}^k = (z_{i_\ell})_{\ell=1}^k, (x_i)_{i \in [N] \setminus I_k} = (z_i)_{i \in [N] \setminus I_k}\big),$$

where we suppose $x_i$ and $y_i$ are sampled independently. These define probability distributions over $[N]^k \times \mathcal{S}^N$. From the independence and Proposition 6, we have

$$\tilde{p}_{\text{moment}}(\boldsymbol{i}, \boldsymbol{z}) = p_{\text{moment}}(\boldsymbol{i}, (z_{i_\ell})_{\ell=1}^k)\mathbb{P}\big((x_i)_{i \in [N] \setminus I_k} = (z_i)_{i \in [N] \setminus I_k}\big)$$
$$= \prod_{\ell=1}^k \frac{p_{i_\ell}(z_{i_\ell})^\beta}{\sum_{i \in [N] \setminus I_{\ell-1}} \|p_i\|_\beta^\beta} \prod_{i \in [N] \setminus I_k} p_i(z_i). \tag{18}$$

Suppose $0 \le \epsilon \le 1$, $\epsilon + \frac{(k-1)S^{1/\alpha}}{N-k+1} < 1$ and recall the set $\mathcal{Z}_\epsilon$ defined in Proposition 5. For $\boldsymbol{z} \in \mathcal{Z}_\epsilon$. From the definition of $\mathcal{Z}_\epsilon$, we have (recall $S = |\mathcal{S}|$)

$$\tilde{p}_{\text{MaskGIT}}(\boldsymbol{i}, \boldsymbol{z}) = \mathbb{P}\big(i_1^* = i_1, \dots, i_k^* = i_k \mid (x_i)_{i=1}^N = (z_i)_{i=1}^N\big) \mathbb{P}\big((x_i)_{i=1}^N = (z_i)_{i=1}^N\big)$$
$$< \left(1 - \epsilon - \frac{(k-1)S^{1/\alpha}}{N-k+1}\right)^{-k} \prod_{\ell=1}^k \frac{p_{i_\ell}(z_{i_\ell})^{1/\alpha}}{\sum_{i \in [N] \setminus I_{\ell-1}} \|p_i\|_\beta^\beta} \prod_{i=1}^N p_i(z_i)$$
$$= \left(1 - \epsilon - \frac{(k-1)S^{1/\alpha}}{N-k+1}\right)^{-k} \prod_{\ell=1}^k \frac{p_{i_\ell}(z_{i_\ell})^{1/\alpha} \cdot p_{i_\ell}(z_{i_\ell})}{\sum_{i \in [N] \setminus I_{\ell-1}} \|p_i\|_\beta^\beta} \prod_{i \in [N] \setminus I_k} p_i(z_i)$$
$$= \left(1 - \epsilon - \frac{(k-1)S^{1/\alpha}}{N-k+1}\right)^{-k} \tilde{p}_{\text{moment}}(\boldsymbol{i}, \boldsymbol{z}), \tag{19}$$

where we have used (18) in the last equality. Let us next bound the total variation distance between $\tilde{p}_{\text{MaskGIT}}$ and $\tilde{p}_{\text{moment}}$. Let us denote $(a)_+ := \max\{0, a\}$. In general, for probability distributions $p$ and $q$ over the same finite set $\mathcal{X}$, we have

$$\sum_{x \in \mathcal{X}} (p(x) - q(x))_+ - \sum_{x \in \mathcal{X}} (q(x) - p(x))_+ = \sum_{x \in \mathcal{X}} (p(x) - q(x)) = 1 - 1 = 0.$$

Thus, for the total variation distance, we have

$$d_{\text{TV}}(p, q) = \frac{1}{2} \sum_{x \in \mathcal{X}} |p(x) - q(x)|$$
$$= \frac{1}{2} \left(\sum_{x \in \mathcal{X}} (p(x) - q(x))_+ + \sum_{x \in \mathcal{X}} (q(x) - p(x))_+\right) = \sum_{x \in \mathcal{X}} (p(x) - q(x))_+.$$

By using this, we have

$d_{\mathrm{TV}}(\tilde{p}_{\mathrm{moment}}, \tilde{p}_{\mathrm{MaskGIT}})$

$= \sum_{\boldsymbol{i} \in [N]^k} \sum_{\boldsymbol{z} \in \mathcal{S}^N} (\tilde{p}_{\mathrm{MaskGIT}}(\boldsymbol{i}, \boldsymbol{z}) - \tilde{p}_{\mathrm{moment}}(\boldsymbol{i}, \boldsymbol{z}))_+$

$= \sum_{\boldsymbol{i} \in [N]^k} \sum_{\boldsymbol{z} \in \mathcal{Z}_\epsilon} (\tilde{p}_{\mathrm{MaskGIT}}(\boldsymbol{i}, \boldsymbol{z}) - \tilde{p}_{\mathrm{moment}}(\boldsymbol{i}, \boldsymbol{z}))_+ + \sum_{\boldsymbol{i} \in [N]^k} \sum_{\boldsymbol{z} \notin \mathcal{Z}_\epsilon} (\tilde{p}_{\mathrm{MaskGIT}}(\boldsymbol{i}, \boldsymbol{z}) - \tilde{p}_{\mathrm{moment}}(\boldsymbol{i}, \boldsymbol{z}))_+$

$< \sum_{\boldsymbol{i} \in [N]^k} \sum_{\boldsymbol{z} \in \mathcal{Z}_\epsilon} \left( \left( 1 - \epsilon - \frac{(k-1)S^{1/\alpha}}{N-k+1} \right)^{-k} - 1 \right) \tilde{p}_{\mathrm{moment}}(\boldsymbol{i}, \boldsymbol{z}) + \sum_{\boldsymbol{i} \in [N]^k} \sum_{\boldsymbol{z} \notin \mathcal{Z}_\epsilon} \tilde{p}_{\mathrm{MaskGIT}}(\boldsymbol{i}, \boldsymbol{z})$

$\leq \left( \left( 1 - \epsilon - \frac{(k-1)S^{1/\alpha}}{N-k+1} \right)^{-k} - 1 \right) + \mathbb{P}\left( (x_i)_{i=1}^N \notin \mathcal{Z}_\epsilon \right). \tag{20}$

By applying Proposition 5, we obtain

$$d_{\mathrm{TV}}(\tilde{p}_{\mathrm{moment}}, \tilde{p}_{\mathrm{MaskGIT}}) \leq \left( \left( 1 - \epsilon - \frac{(k-1)S^{1/\alpha}}{N-k+1} \right)^{-k} - 1 \right) + \exp\left( -\frac{3}{8} \epsilon^2 N S^{-1/\alpha} \right).$$

For $0 < \delta \leq \frac{1}{2k}$, we have

$$(1-\delta)^{-1} = 1 + \delta \sum_{n=0}^{\infty} \delta^n \leq 1 + \frac{\delta}{1 - \frac{1}{2k}} = 1 + \frac{2k\delta}{2k-1}.$$

Since $(k-1)\frac{2k\delta}{2k-1} \leq \frac{k-1}{2k-1} \leq 1/2$, we have

$$(1-\delta)^{-k} \leq \left( 1 + \frac{2k\delta}{2k-1} \right)^k = 1 + \sum_{n=1}^{k} \binom{k}{n} \left( \frac{2k\delta}{2k-1} \right)^n$$

$$\leq 1 + k \cdot \frac{2k\delta}{2k-1} \sum_{n=1}^{k} \frac{(k-1)^{n-1}}{n!} \left( \frac{2k\delta}{2k-1} \right)^{n-1}$$

$$\leq 1 + k \cdot \frac{2k\delta}{2k-1} \sum_{m=0}^{\infty} \frac{1}{m!} \left( (k-1)\frac{2k\delta}{2k-1} \right)^m$$

$$= 1 + \frac{2k^2\sqrt{e}}{2k-1} \delta < 1 + 4k\delta.$$

Thus, assuming $\epsilon + \frac{(k-1)S^{1/\alpha}}{N-k+1} < \frac{1}{4k}$, we have

$$d_{\mathrm{TV}}(\tilde{p}_{\mathrm{moment}}, \tilde{p}_{\mathrm{MaskGIT}}) \leq 4k \left( \epsilon + \frac{(k-1)S^{1/\alpha}}{N-k+1} \right) + \exp\left( -\frac{3}{8} \epsilon^2 N S^{-1/\alpha} \right). \tag{21}$$

Since the total variation distance is always bounded by 1, actually (21) holds without the posed assumptions on $\epsilon + \frac{(k-1)S^{1/\alpha}}{N-k+1}$.

Note that, when $N < k^2 |\mathcal{S}|^{1/\alpha}$, the upper bound of $d_{\mathrm{TV}}(p_{\mathrm{moment}}, p_{\mathrm{MaskGIT}})$ becomes larger than 1, which holds trivially true since $d_{\mathrm{TV}}$ is bounded above by 1. So, it suffices to prove the desired inequality with $\log$ instead of $\log^+$ under the assumption $N \geq k^2 |\mathcal{S}|^{1/\alpha}$. Under this, by letting $\epsilon = \sqrt{\frac{8}{3} \frac{S^{1/\alpha}}{N} \cdot \frac{1}{2} \log \frac{N}{k^2 S^{1/\alpha}}}$ in (21), we have

$d_{\mathrm{TV}}(\tilde{p}_{\mathrm{moment}}, \tilde{p}_{\mathrm{MaskGIT}})$

$$\leq 4k \sqrt{\frac{4}{3} \frac{S^{1/\alpha}}{N} \log \frac{N}{k^2 S^{1/\alpha}}} + \frac{4k(k-1)S^{1/\alpha}}{N-k+1} + \sqrt{\frac{k^2 S^{1/\alpha}}{N}}. \tag{22}$$

We need to make sure $\epsilon \leq 1$, but the above bound is valid even when $\epsilon > 1$, again because of the boundedness of $d_{\mathrm{TV}}$. From $N \geq k^2 |\mathcal{S}|^{1/\alpha} \geq k^2$, we also have

$$\frac{4k(k-1)S^{1/\alpha}}{N-k+1} \leq 4S^{1/\alpha} \cdot \frac{k^2-k}{N-k} \leq 4S^{1/\alpha} \cdot \frac{k^2}{N} = \frac{4k^2 S^{1/\alpha}}{N} \leq 4\sqrt{\frac{k^2 S^{1/\alpha}}{N}}.$$

By applying this to (22), we obtain

$$
\begin{aligned}
d_{\mathrm{TV}}(\tilde{p}_{\mathrm{moment}}, \tilde{p}_{\mathrm{MaskGIT}}) &\leq 4k\sqrt{\frac{4}{3}\frac{S^{1/\alpha}}{N}\log\frac{N}{k^2 S^{1/\alpha}}} + 5\sqrt{\frac{k^2 S^{1/\alpha}}{N}} \\
&= \sqrt{\frac{k^2 S^{1/\alpha}}{N}}\left(5 + \sqrt{\frac{64}{3}\log\frac{N}{k^2 S^{1/\alpha}}}\right) \\
&\leq 5\sqrt{\frac{k^2 S^{1/\alpha}}{N}}\left(1 + \sqrt{\log\frac{N}{k^2 S^{1/\alpha}}}\right).
\end{aligned}
\tag{23}
$$

Finally, by denoting $I = \{i_1, \dots, i_k\}$, we have

$$
d_{\mathrm{TV}}(p_{\mathrm{moment}}, p_{\mathrm{MaskGIT}})
$$

$$
= \frac{1}{2}\sum_{\boldsymbol{i}\in[N]^k}\sum_{(z_i)_{i\in I}\in\mathcal{S}^k}|p_{\mathrm{MaskGIT}}(\boldsymbol{i},(z_i)_{i\in I}) - p_{\mathrm{moment}}(\boldsymbol{i},(z_i)_{i\in I})|
$$

$$
= \frac{1}{2}\sum_{\boldsymbol{i}\in[N]^k}\sum_{(z_i)_{i\in I}\in\mathcal{S}^k}\left|\sum_{(z_j)_{j\notin I}\in\mathcal{S}^{N-k}}(p_{\mathrm{MaskGIT}}(\boldsymbol{i},(z_i)_{i\in I},(z_j)_{j\notin I}) - p_{\mathrm{moment}}(\boldsymbol{i},(z_i)_{i\in I},(z_j)_{j\notin I}))\right|
$$

$$
\leq \frac{1}{2}\sum_{\boldsymbol{i}\in[N]^k}\sum_{(z_i)_{i\in I}\in\mathcal{S}^k}\sum_{(z_j)_{j\notin I}\in\mathcal{S}^{N-k}}|p_{\mathrm{MaskGIT}}(\boldsymbol{i},(z_i)_{i\in I},(z_j)_{j\notin I}) - p_{\mathrm{moment}}(\boldsymbol{i},(z_i)_{i\in I},(z_j)_{j\notin I}))|
$$

$$
= d_{\mathrm{TV}}(\tilde{p}_{\mathrm{moment}}, \tilde{p}_{\mathrm{MaskGIT}}).
$$

By combining this with (23), we obtain the desired conclusion. □

## D  ADDITIONAL EXPERIMENTAL DETAILS

### D.1  SAMPLING SCHEDULE

Let $D$ be the number of positions (so $\boldsymbol{x}\in\mathcal{S}^D$) and $N$ be the number of total sampling steps. Let $J_n \subset [D]$ be the (random) set of indices that are open after the $n$-th step, i.e., $\emptyset = J_0 \subset J_1 \subset \cdots \subset J_N = [D]$. Let $I_n := J_n \setminus J_{n-1}$ for $n = 1, \dots, N$, which is the (random) set of indices we unmask at the $n$-th step. In all the experiments, the cardinalities $|J_n|$ and $|I_n|$ $(= |J_n| - |J_{n-1}|)$ are predetermined by the unmasking size schedule such as:

- Cosine schedule: $|J_n| = \mathrm{round}(\cos(\frac{\pi}{2}D(1 - \frac{n}{N})))$.
- Uniform schedule: $|J_n| = \mathrm{round}(D \cdot \frac{n}{N})$.

Here, "round" means integer rounding. We adopted the cosine schedule for image and the uniform schedule for language.

Let us now consider specifically the $n$-th sampling step out of $N$ steps. Given the denoising model $p$, we use the marginal distributions $(p_{i|J_{n-1}})_{i\in[D]\setminus J_{n-1}}$ for this step. Let $k$ be the number of indices to unmask in this step, determined by the unmasking size schedule. Then, for the sampler **MaskGIT**, we use $\mathrm{OneRoundMaskGIT}((p_{i|J_{n-1}})_{i\in[D]\setminus J_{n-1}}, k, \alpha_n)$ from Algorithm 1 to determine $I_n$ and $\boldsymbol{x}_{I_n}$. Here, $\alpha_n$ is the Gumbel temperature for the $n$-th step, which is scheduled as $\alpha_n = \alpha(1 - n/N)$ following (Chang et al., 2022), where $\alpha$ is the temperature parameter of the method presented in the figures (e.g., Figure 3). We use the same $\alpha_n$ for **Moment** (Algorithm 2) and its variants given the parameter $\alpha$. Note that, in the final step ($n = N$) of **Moment** or other temperature-sampling methods, we omit the sampling temperature (or take $\alpha_N \to \infty$), in order that it corresponds to the final step of **MaskGIT**.

### D.2  PARTIAL CACHING

In the partial caching algorithm we described in Section 4.1, we have a degree of freedom in dividing the selected index set $I$ into $A$ and $B$ (where we have $A \cup B = I$ and $A \cap B = \emptyset$).

Let us explain our implementation. Let us use the notation of $J_n$ and $I_n$ introduced in the previous section. In the $n$-th step, suppose we decompose $I_n$ into $A_n$ and $B_n$, where $A_n$ is the set of indices unmasked in the intermediate step of partial caching. If we let $J_{n-1/2} := J_{n-1} \cup A_n$ for $n \geq 1$, then we adopted the canonical extension of the scheduler in Appendix D.1:

- Cosine schedule: $|J_{n-1/2}| = \text{round}(\cos(\frac{\pi}{2} D(1 - \frac{n-1/2}{N})))$.

- Uniform schedule: $|J_{n-1/2}| = \text{round}(D \cdot \frac{n-1/2}{N})$.

Thus, the cardinality of $A_n$ was determined by $|A_n| = |J_{n-1/2}| - |J_{n-1}|$, depending on the sampling schedule we use. Since each of our sampling algorithm outputs an ordering of masked indices (from which we determine $I_n$), we simply determine $A_n$ as the top-$k$ of the ordered indices ($k = |A_n| = |J_{n-1/2}| - |J_{n-1}|$).

### D.2.1 COMPUTATIONAL EFFICIENCY OF PARTIAL CACHING

The total cost of *attention computation* in partial caching is $1 + |I|/D$ times the original full attention computation (in terms of the notations in Section 4.1). However, it costs more computation in caching the vectors and other CPU/GPU operations. Indeed, while caching shows some performance gain with A6000 latency in the language experiment (Figure 5, Right), it vanishes when we use H100 GPU (Figure 6, Right). It would be caused by the faster attention computation of H100, reducing its weight among the overall computational cost and making other computational overhead apparent.

### D.3 ADDITIONAL DETAILS ON IMAGE MODELING EXPERIMENTS

The MAGE ViT-B model (Li et al., 2023), which we used in the experiments, can be regarded as a masked diffusion model on the space of a pretrained VQGAN tokenizers (Esser et al., 2021). It was trained on the ImageNet $256 \times 256$ dataset (Deng et al., 2009). The codebook size is given by $|\mathcal{S}| = 1024$, and the length of each token sequence (corresponding to a single image) is $D = 256$. Each experiment with MAGE was conducted with a single A6000 GPU with a minibatch size of 64. Based on 50000 unconditional images generated from each sampler, we measured FID and IS against the ImageNet dataset by using **torch-fidelity**[2], following the description of the repository of MAGE[3].

### D.4 ADDITIONAL DETAILS ON LANGUAGE MODELING EXPERIMENTS

We used the SDTT model (Deschenaux & Gulcehre, 2025)[4], which is a masked diffusion model over a GPT-2 tokenizers (Radford et al., 2019). It was trained on the OpenWebText dataset (Gokaslan & Cohen, 2019). The codebook size is $|\mathcal{S}| = 50257$ and the token sequence length is given by $D = 1024$. Most experiments were conducted on a single H100 GPU, while the preliminary experiments, the ones in Appendix D.4.1, and the latency computation in Figure 4(Right) were conducted on a single A6000 GPU. Each plot was computed by 1024 samples generated with a minibatch size of 16, using the following performance metrics.

**Generative Perplexity.** It was measured against the GPT-2 large model (Radford et al., 2019) and averaged over 1024 samples. We used the implementation of Deschenaux & Gulcehre (2025).

**Entropy.** Following the existing work (Gat et al., 2024; Zheng et al., 2025), we measured the sentence-wise entropy for checking the diversity of generated sentences. In our implementation (following the description of (Zheng et al., 2025)), for a sequence of tokens $\boldsymbol{x} = (x_1, \ldots, x_D)$, we define the sentence entropy as

$$- \sum_{s \in \mathcal{S} \cap \boldsymbol{x}} \frac{\#\{i \in [D] \mid x_i = s\}}{D} \log \frac{\#\{i \in [D] \mid x_i = s\}}{D},$$

where $\mathcal{S} \cap \boldsymbol{x}$ is the set of tokens appearing in $\boldsymbol{x}$. Its average over 1024 samples was plotted.

---

[2]https://github.com/toshas/torch-fidelity.
[3]https://github.com/LTH14/mage.
[4]It was loaded by load_small_student(loss="kld", round=7) in the repository https://github.com/jdeschena/sdtt.

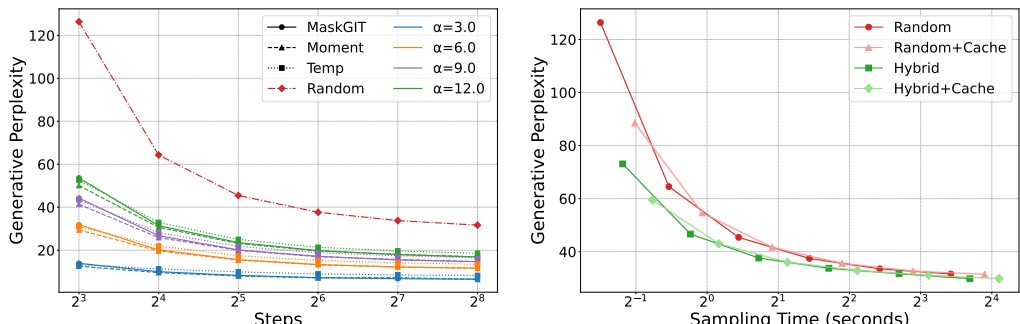

Figure 6: Additional experimental results. (*Left*) Generative Perplexity of various samplers with temperature sampling. (*Right*) Generative Perplexity of our proposed samplers against sampling time per batch on H100 GPU.

Figure 6(Left) shows the omitted results for Generative Perplexity. While the methods with lower temperature apparently attain better the generation quality, lowering the temperature extremely harms the diversity in reality (Figure 5, Left).

### D.4.1 NUMERICAL PRECISION

Zheng et al. (2025) pointed out that the sampling from masked diffusion models with low numerical precision (32-bit) can lead to errors in categorical sampling, showing lower (better) Generative Perplexity at the cost of lower (worse) Entropy. Because of this, they suggest using 64-bit computation for a fair evaluation of masked diffusion models.

However, as they also note in Section J.2.2 of their paper, it is primarily because the lower numerical precision results in biased positional selection rather than the sampling distribution shift at each position. In our case, since we fix the token positions in most samplers (except **MaskGIT**), we do not suffer from this problem though we use 32-bit precision. Indeed, Table 1 shows that, while the vanilla sampler in discrete diffusion suffer from the difference in numerical precision, **Fixed**, which corresponds to **Random** in the main text, exhibits similar results in both precision settings.

Table 1: Comparison of different numerical precision in **Vanilla** and **Fixed** samplers. **Vanilla** is a standard sampler for discrete diffusion, where it independently determines whether or not unmasking a certain position. **Fixed** is a sampler that pre-determines the number of unmasked positions at each step, and it determines which positions to unmask uniformly at random. Both follows the uniform schedule (Appendix D.1) in expectation.

| Sampler | Precision | 8 steps | | 32 steps | | 128 steps | |
|---|---|---|---|---|---|---|---|
| | | Gen. PPL | Entropy | Gen. PPL | Entropy | Gen. PPL | Entropy |
| **Vanilla** | 32-bit | 125.62 | 5.40 | 41.91 | 5.31 | 27.63 | 5.17 |
| | 64-bit | 137.95 | 5.42 | 46.57 | 5.35 | 33.80 | 5.28 |
| **Fixed** | 32-bit | 131.01 | 5.41 | 45.10 | 5.32 | 33.15 | 5.26 |
| (= **Random**) | 64-bit | 130.76 | 5.41 | 46.66 | 5.35 | 34.29 | 5.29 |

### D.4.2 **Hybrid** ALGORITHM

Let us explain the details of the **Hybrid** sampler in the language experiments, where we merged the **Halton** and **U-Moment** samplers. Let us consider the $n$-th sampling step out of $N$ total steps and let $J_{n-1}$ be the set of indices already unmasked at this stage. The ordering of first $k = |I_n|$ (where $I_n = J_n \setminus J_{n-1}$ is from Appendix D.1) positions from each sampler is given as follows:

- **Halton**: We consider the one-dimensional Halton sequence of indices (with base 2), i.e., rearrangement of $[D]$, and let $\boldsymbol{i} = (i_1, \ldots, i_k)$ be its first $k$ entries that are also in $[D] \setminus J_{n-1}$.

- **U-Moment**: As in Algorithm 2, we define the ordering $\boldsymbol{j} = (j_1, \ldots, j_k)$ by

$$
\boldsymbol{j} = \operatorname{argtop} k_{j \in [D] \setminus J_{n-1}} \left\{ \log \sum_{x \in \mathcal{S}} p_{j|J_{n-1}}(x|\boldsymbol{x}_{J_{n-1}})^{\beta} + \xi_j \right\},
$$

where the exponent $\beta = 1 + 1/\alpha$ is determined by the temperature parameter $\alpha$, and $\xi_j$ is an independently sampled standard Gumbel noise for each $j \in [D] \setminus J_{n-1}$.

We then merge $\boldsymbol{i}$ and $\boldsymbol{j}$ into $\boldsymbol{k}$ as described in Section 4.2 to obtain an ordering for **Hybrid**, where merging parameter $m = m_n$, controlling how many indices we take from $\boldsymbol{i}$, is scheduled as $m_n = \operatorname{round}((1-n/N)|I_n|)$. Intuitively, it means that we basically start from **Halton**, whose exploration works at the initial stages, and gradually move to **U-Moment**, which conducts exploitation-based index selection. In the implementation of **Hybrid+Cache**, we just apply the caching procedure as explained in Appendix D.2 to the above merged ordering $\boldsymbol{k}$.

# E  FURTHER EXPERIMENTAL RESULTS

## E.1  UNCONDITIONAL IMAGE GENERATION WITH MAGE

In addition to the raw data presented in Figure 3, we give the detailed FID values of **Random**, **MaskGIT**, **Moment**, **Temp** at each temperature in Table B. Based on these numbers, we also present the relative differences between **MaskGIT** and the other samplers in Table A in the main body, which validates our theory on **Moment** approximating **MaskGIT** with the same temperature.

Table B: ImageNet FIDs for various methods in unconditional experiment with MAGE.

| Steps | Random | MaskGIT | | | Moment | | | Temp | | |
|---|---|---|---|---|---|---|---|---|---|---|
| | | $\alpha = 3.0$ | 6.0 | 12.0 | 3.0 | 6.0 | 12.0 | 3.0 | 6.0 | 12.0 |
| 8 | 49.86 | 18.95 | 19.21 | 27.18 | 19.47 | 17.42 | 24.26 | 20.83 | 18.74 | 25.26 |
| 16 | 31.85 | 19.36 | 11.29 | 13.32 | 20.71 | 11.52 | 12.73 | 19.86 | 12.09 | 13.25 |
| 32 | 22.55 | 24.61 | 11.45 | 9.84 | 25.76 | 11.77 | 9.73 | 23.83 | 11.73 | 10.00 |
| 64 | 18.40 | 27.23 | 12.36 | 9.03 | 27.80 | 12.77 | 9.13 | 25.32 | 12.33 | 9.27 |

## E.2  CLASS-CONDITIONAL IMAGE GENERATION WITH MASKGIT-PYTORCH

To confirm the approximation ability of **Moment** in conditional image generation, we used the MaskGIT-PyTorch (Besnier & Chen, 2023) model pretrained on ImageNet $256 \times 256$. We adopted the arccos sampling scheduler (see also Appendix D.1) for unmasking size, and the Gumbel temperature schedule followed the default implementation (same as that in Appendix D.1). The classifier-free guidance coefficient was set to 3.0 throughout the experiment.

Table C shows the FID values of several samplers and Table D shows their relative differences to **MaskGIT** samplers (corresponding to Table B and Table A in the unconditional experiment, respectively). We can confirm again that **Moment** samplers consistently achieve the FID within 10% with **MaskGIT** of corresponding temperature. This approximation can also be visibly confirmed by the plot in Figure C.

## E.3  LANGUAGE EXPERIMENTS WITH DIFFERENT MODELS

To obtain further empirical evidence of the efficiency of our methods in Section 4, we additionally tested them against a larger model (Deschenaux & Gulcehre, 2025, which we refer to as SDTT-large) and a model with a modified architecture (Hayakawa et al., 2025, which we refer to as Di4C). Each datapoint is based on 256 generated sequences of 1024 tokens, with 8 to 256 sampling steps. In each experiment, **Hybrid** used Gumbel coefficient of $\alpha = 6.0$ as in the main body.

Table C: FIDs for various methods and steps in class-conditional experiment with MaskGIT-PyTorch.

| Steps | Random | **MaskGIT** | | | **Moment** | | | **Temp** | | |
|---|---|---|---|---|---|---|---|---|---|---|
| | | $\alpha = 3.0$ | 6.0 | 12.0 | 3.0 | 6.0 | 12.0 | 3.0 | 6.0 | 12.0 |
| 4 | 30.11 | 14.34 | 19.17 | 23.83 | 12.85 | 17.92 | 23.01 | 17.65 | 22.82 | 25.97 |
| 8 | 12.92 | 6.96 | 6.60 | 7.67 | 7.26 | 6.41 | 7.21 | 7.02 | 8.20 | 9.89 |
| 16 | 7.85 | 8.65 | 6.79 | 6.08 | 9.43 | 7.39 | 6.12 | 7.24 | 6.18 | 6.41 |
| 32 | 6.37 | 10.28 | 8.10 | 6.72 | 11.15 | 8.94 | 7.09 | 8.43 | 6.45 | 5.91 |

Table D: Mean relative difference (%) of ImgeNet FIDs against reference MaskGIT samplers across 4, 8, 16, 32 steps in class-conditional experiment with MaskGIT-PyTorch. Mean relative difference was computed in the same way as Table A. Bolded under $10\%$. **Moment** consistently approximates **MaskGIT** with the same temperature.

| Ref. **MaskGIT** temperature | Random | **MaskGIT** | | | **Moment** | | | **Temp** | | |
|---|---|---|---|---|---|---|---|---|---|---|
| | | $\alpha = 3.0$ | 6.0 | 12.0 | 3.0 | 6.0 | 12.0 | 3.0 | 6.0 | 12.0 |
| $\alpha = 3.0$ | 60.7 | - | 20.4 | 35.2 | **8.1** | 15.1 | 31.1 | 14.5 | 35.7 | 47.9 |
| $\alpha = 6.0$ | 47.5 | 21.3 | - | 17.0 | 29.9 | **7.1** | 12.9 | **6.3** | 18.2 | 29.5 |
| $\alpha = 12.0$ | 32.4 | 36.1 | 16.4 | - | 43.1 | 23.9 | **3.9** | 19.7 | **4.2** | 13.9 |

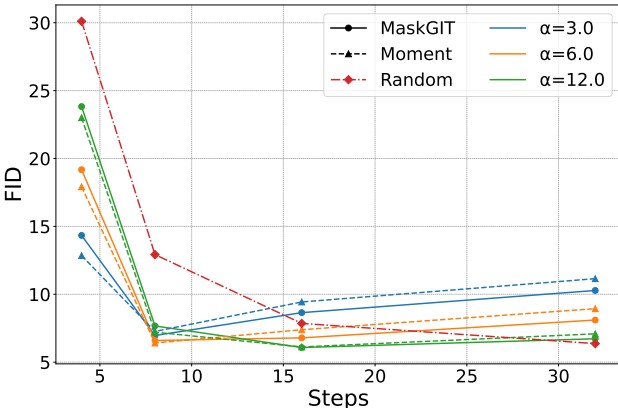

Figure C: FID against number of steps for various samplers with MaskGIT-PyTorch. FID was computed with 50,000 (50 per class) class-conditional generation results.

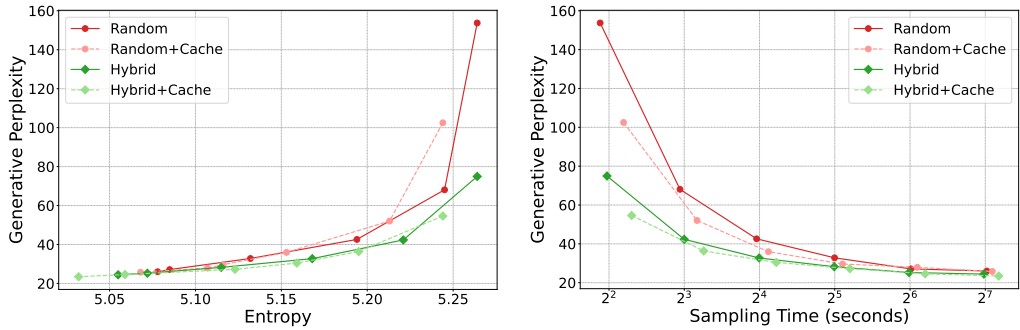

Figure D: Experiments with SDTT-large. Generative Perplexity was measured by Llama3-8B. (*Left*) Trade-off between Generative Perplexity and Entropy, (*Right*) Generative Perplexity against Sampling Time on A6000.

**SDTT-large model with Hybrid and Partial caching.** In the experiment with a larger model, we used SDTT-large, a pretrained transformer model scaled to 863M parameters, in contrast to SDTT with 169M parameters used in Section 5.2. Since SDTT-large is larger than the GPT-2 large model (774M parameters) in scale, we used Llama3-8B (Dubey et al., 2024) for computing generative perplexities.

Figure D shows the results for SDTT-large. We can see a similar tradeoff improvement of **Hybrid** over **Random** and consistent speedups of **Hybrid** and **Random+Cache** over **Random**. We also plotted a comparison of SDTT-large and SDTT with **Random** and **Hybrid** samplers in Figure E. While the model difference becomes dominant as we increase sampling steps, the use of **Hybrid** is effective for moth models, especially in the few-step regime.

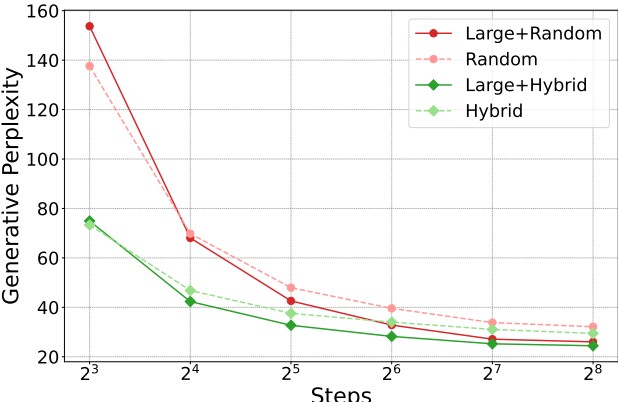

Figure E: Efficiency comparison of SDTT-large and SDTT with **Random** and **Hybrid** samplers. 'Large+' means SDTT-large; otherwise it shows the results of SDTT. Generative Perplexity was measured by Llama3-8B for both models.

**Hybrid sampler with mixture modeling.** We applied **Hybrid** to Di4C, a model obtained from SDTT after two rounds of Di4C-finetuing (Hayakawa et al., 2025)[5]. This Di4C finetuning includes a slight model architecture modification, which adds a random latent variable to capture dimensional correlations between different positions/tokens (caching is not applicable because of this modified modeling). Figure F shows the comparison including SDTT results. Combining **Hybrid** and the Di4C finetuning (**Di4C+Hybrid**) sums up individual performance gains of **Hybrid** and Di4C from the original result of **Random**; this is especially remarkable in the few-step regime.

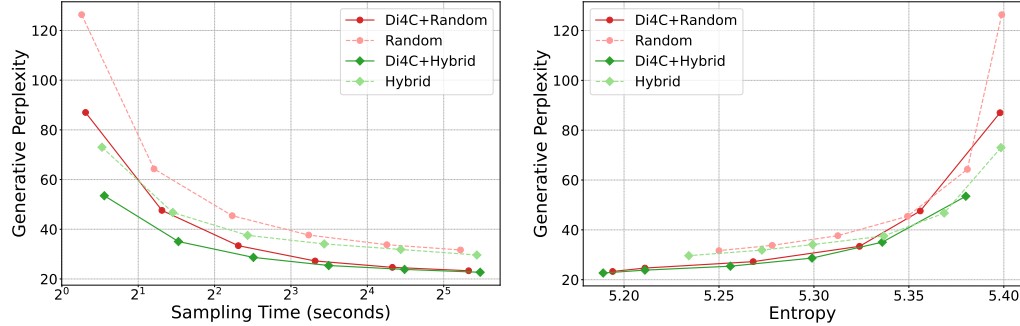

Figure F: Efficiency comparison of Di4C and SDTT with **Random** and **Hybrid** samplers. Plots without 'Di4C+' show the results of SDTT. Generative Perplexity was measured by GPT-2 large for both models.

---

[5] `sdtt7-di4c2.ckpt` in `https://zenodo.org/records/15124163`

## E.4 ASYMMETRY OF HYBRID MODELING

**Hybrid** is asymmetric regarding the order of two merged methods as detailed in Appendix D.4.2, where **Halton** determines the indices to unmask and then **U-Moment** determines the rest of unmasking. Thus, we can also consider **s-Hybrid**, where we just swap the order of two methods in each step, while the number of indices allocated to each method is unchanged. We show the results of **s-Hybrid** (256 samples per datapoint) with SDTT among other samplers in Table E. While **s-Hybrid** shows slightly worse performance compared to **Hybrid**, the overall behavior is similar, especially in the few-step regime. The potential source of the difference is that, since the Halton sequence is deterministic, its low-discrepancy effect can be reduced when we already select some positions. To mitigate this effect, adaptive methods such as determinantal point processes (Kulesza et al., 2012) as a replacement for **Halton** can be promising in future work.

Table E: Generative Perplexity and Entropy of **f-Hybrid** and other samplers applied to SDTT.

| Metric | Method | Steps | | | | | |
|--------|--------|-------|-------|-------|-------|-------|-------|
| | | 8 | 16 | 32 | 64 | 128 | 256 |
| Gen. PPL | **Random** | 126.36 | 64.31 | 45.45 | 37.67 | 33.78 | 31.67 |
| | **U-Moment** | 107.04 | 55.36 | 38.95 | 31.96 | 28.40 | 26.62 |
| | **Hybrid** | 73.02 | 46.73 | 37.55 | 34.09 | 31.87 | 29.64 |
| | **s-Hybrid** | 73.46 | 48.76 | 40.25 | 35.30 | 31.97 | 30.84 |
| | **Halton** | 55.09 | 42.53 | 37.41 | 35.76 | 34.32 | 33.13 |
| Entropy | **Random** | 5.399 | 5.381 | 5.349 | 5.313 | 5.278 | 5.250 |
| | **U-Moment** | 5.354 | 5.333 | 5.298 | 5.263 | 5.228 | 5.199 |
| | **Hybrid** | 5.399 | 5.369 | 5.337 | 5.299 | 5.273 | 5.234 |
| | **s-Hybrid** | 5.382 | 5.344 | 5.318 | 5.278 | 5.229 | 5.198 |
| | **Halton** | 5.426 | 5.385 | 5.351 | 5.320 | 5.293 | 5.263 |

## E.5 GENERATED SAMPLES

### E.5.1 MAGE WITH **MaskGIT** AND **Moment** SAMPLERS

We plot the generated samples of MAGE using these samplers with 16 steps and 64 steps in Figure G. Note that, although they are generated from the same seed, samples do not correspond to each other between **MaskGIT** and **Moment** because of the different uses of random numbers as in Figures A and B.

### E.5.2 SDTT WITH **MaskGIT** AND **Hybrid** SAMPLERS

To see the actual examples of entropy reduction and its mitigation by our methods, we put the 8-step generation results of **MaskGIT** and **Hybrid** with $\alpha = 3.0, 6.0$. We can see the repetitive use of the same characters in **MaskGIT** samplers, which harms the Entropy, while **Hybrid** results look more natural with diverse use of words, even with higher (worse) Generative Perplexities.

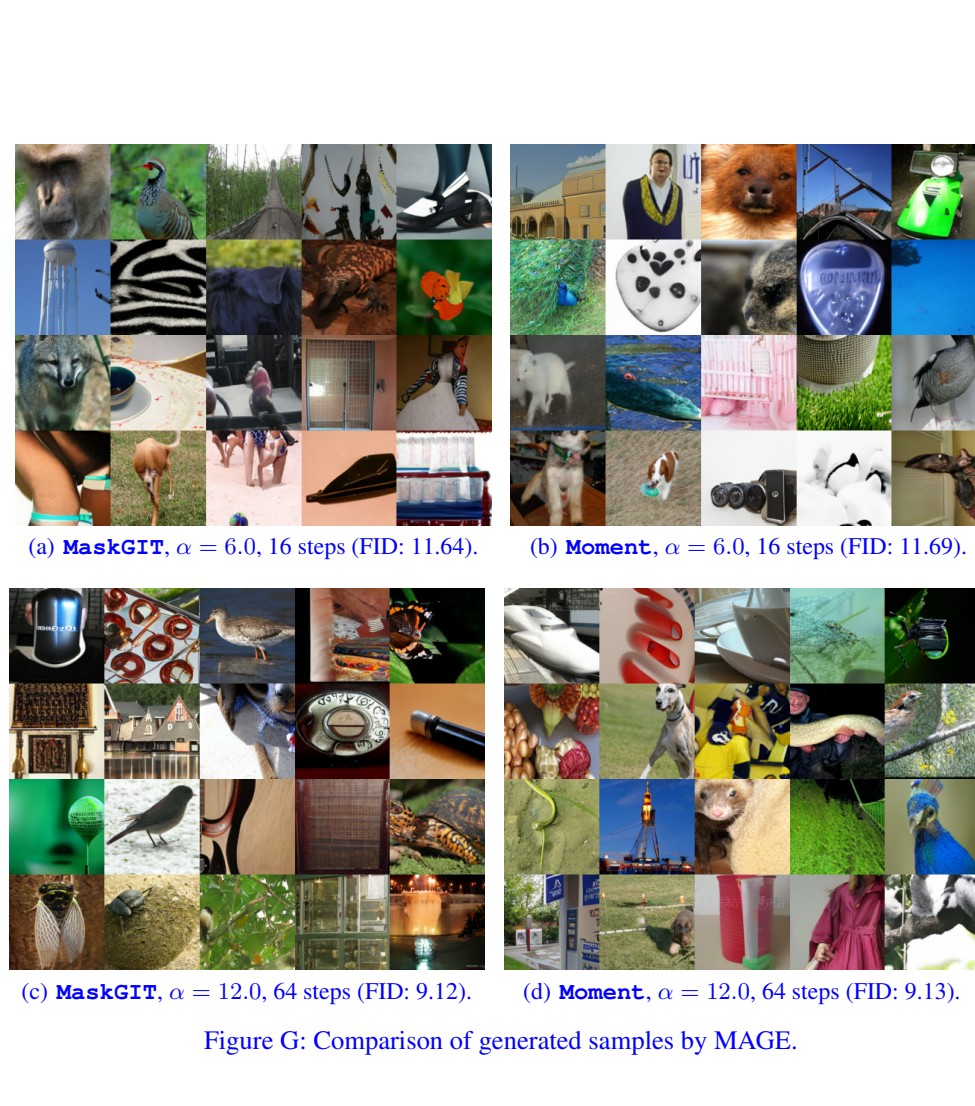

(a) **MaskGIT**, $\alpha = 6.0$, 16 steps (FID: 11.64).      (b) **Moment**, $\alpha = 6.0$, 16 steps (FID: 11.69).

(c) **MaskGIT**, $\alpha = 12.0$, 64 steps (FID: 9.12).      (d) **Moment**, $\alpha = 12.0$, 64 steps (FID: 9.13).

Figure G: Comparison of generated samples by MAGE.

<|endoftext|>iz and for free speech. And to I. however, the other night, and to 'indicate' his comments, he argued that I. of the.e. did not say, and to prove to the woman he did not speak the following, and that it is not true.

I. he said that, that and that. and that I. and I." and, that "I have. and, I. and, in the case of I. and. and. and her,.. and and. and her,," and he reports that the public has been framed in a manner that is reported to him.

I. he connected I. to the communication of the.e to the.e, and to the case of the.e. and. with the case of I. in the. and. I. and I. II, and, I., and,, that was the case of the.e. and, did not decide, I. and. and I. and. and. in the case of I. in the. and. I." he about a, that I. I, and. I, and I., and I

Figure H: Generated by SDTT with **MaskGIT**, $\alpha = 3.0$ (Gen. PPL: 13.88, Entropy: 3.320).

<|endoftext|> and get to show off your help. Don't worry about that side of the screen, the beta version gets released on June 5, 2015.
[U+2620]:
What's available here is a free download set up for a variety of machines, naturally; specifically to vDB, MooseX, and the UE6 Server. It is prided itself on the recent release of a new GUI server, and the game is well over there, as there are yet to play too.
Unofficial demos, which focuses on the development, does not release on CODAs as Saturday 3, 2015.
The the current release is the server release. The New Descripten was created late and released in 2012, and several developers have since been involved, working on various different games over a year or so.
In addition, the community has already begun out beta testing on the server and client in an attempt to polish them and ability to show them through the client publicly. Once that can play out, potential releases kaizen out this summer, and beyond this fall, it is enough time for the human capital of release to remain in place. To ensure the delay, however, a group of senior officials, Republicans, and

Figure I: Generated by SDTT with **Hybrid**, $\alpha = 3.0$ (Gen. PPL: 67.98, Entropy: 5.360). [U+2620] is actually a single Unicode character of crossbones.

<|endoftext|> and unworthy of free speech. According to Judge Sitton, the FBI, had listened to respond to said individuals' their comments, and argued against it. But the judge,..t., refused to respond to the motion and did not respond, on the grounds that it is not confidentiality.
"The FBI does not, and doesn.. have a right stopping this," he wrote. "This matter, therefore, can't be mentioned in the reported briefs."
However,..t. wrote that "there are a number of reports that appear to have been contained in the case that turns up. .
.
So, perhaps considered the possibility of taking action on. or on that, and reviewing it.
While this was in time with the.A. and.A. it occurred to me that Mr.. A. v. A. is an important part of the history of the FBI and, until now, not very much known. I had talked to Mr. Sullivan, and in a conversation with ". Barry, the AFR," he about a year ago, indicated today, and today is about 10:30 to 11:

Figure J: Generated by SDTT with **MaskGIT**, $\alpha = 6.0$ (Gen. PPL: 31.89, Entropy: 4.465).

<|endoftext|> and get a show of your game. Don't worry about right side of the screen, the physical version gets certified on April 5, 2015.
Now read:
We've read here about a game download set to ship a perfect three months, offering various changes to game quality in Moosehead, by the time PMabouts. Microsoft has Guided on the most recent release of the new development game, and that product is well too available, as we are excited to test some.
The gameplay demos will be shown on Steam, which does not release on CMA1 as Saturday, October 12.
The the ship will be the first release of The Invaders, Raiden, the Ocean and Star Fox games, and will ship on the same date, but on a different console about a year after arrival.
So far, six games have already been confirmed for testing on this ship, and we will attempt to get begin to publicly ship with them during the summer (because the year can be a difficult time to try them out). Please thank you for your game, as soon as it arrives in our capital, bear in mind each other. To get the latest information and set your sights on at least, you can find

Figure K: Generated by SDTT with **Hybrid**, $\alpha = 6.0$ (Gen. PPL: 73.02, Entropy: 5.399).

