# OpenReview forum: "Demystifying MaskGIT Sampler and Beyond: Adaptive Order Selection in Masked Diffusion"
_ICLR.cc/2026/Conference — Submitted to ICLR 2026_

### Official Review · Reviewer_Xexd · 2025-10-28

**Soundness:** 2
**Presentation:** 2
**Contribution:** 2
**Rating:** 4
**Confidence:** 3

**Summary:**

This paper provides a deep analysis of the MaskGIT sampler's behavior in image and text generation, revealing its implicit temperature sampling mechanism and explaining its degraded performance with increased sampling steps. Building on this theoretical analysis, the authors introduce the "moment sampler," an asymptotically equivalent but more tractable and interpretable alternative that employs a "choose-then-sample" (CTS) strategy. To further enhance the efficiency of CTS algorithms, two key techniques are proposed:
1）Partial Caching Approximation: For Transformer-based models, this technique approximates sampling trajectories with more effective steps without proportionally increasing computational cost.
2）Hybrid Approach for Exploration-Exploitation Balancing: This formalizes the exploration-exploitation trade-off in adaptive unmasking for masked diffusion sampling, developing a hybrid method that combines the strengths of both.
Experiments on the ImageNet image dataset and OpenWebText text dataset validate the theoretical findings and demonstrate the efficiency and effectiveness of the proposed methods.

**Strengths:**

1. Theoretical Analysis. The paper provides the theoretical analysis of the MaskGIT sampler, uncovering its implicit temperature sampling mechanism and explaining its performance degradation with increased sampling steps. This understanding is crucial for further advancements in masked diffusion models.
2. The introduction of the moment sampler offers an asymptotically equivalent yet more interpretable and tractable alternative. It adopts "choose-then-sample" strategy and proposes partial caching approximation and the hybrid approach for exploration-exploitation balancing. Partial caching promises to improve sampling efficiency without significantly increasing computational cost, while the hybrid method effectively balances exploration and exploitation in sampling strategies.

**Weaknesses:**

1. It is a critical problem to determine the tokens to be sampled in the "select-and-sample" paradigm. The analyses in this part are insufficient. It confuses me how Algorithm 2 is implemented and how to determine the k tokens to be sampled. More details and maybe code are encouraged to verify the actual modification of this method over Maskgit.
2. This paper employs the unconditional MAGE as the baseline, lacking necessary verification on diverse models, e.g., the class-conditional Maskgit, and text-conditional model.
3. Even if on MAGE, the performance gain is marginal compared to the baseline as shown in Figure 3 and 4. Besides, a quantitative performance table are encouraged for better visualization.

**Questions:**

While the partial caching method aims for efficiency, the paper notes that its performance improvement is not significant on certain GPUs (e.g., H100 GPU), and computational overhead can even become noticeable (Section D.2.1). This may suggests that the practical benefits of this method might be highly dependent on hardware and model architecture, and its general applicability requires further investigation.

---

> ### Author Response · Authors · 2025-11-21
> **Rebuttal reply to Reviewer Xexd**
>
> We are deeply grateful to your comment that our theoretical understanding is "crucial for further advancements in masked diffusion models." Let us further answer your comments and questions.
>
> ### [X-1] Regarding token selection in "choose-then-sample" paradigm
>
> > It is a critical problem to determine the tokens to be sampled in the "select-and-sample" paradigm. The analyses in this part are insufficient. It confuses me how Algorithm 2 is implemented and how to determine the k tokens to be sampled. More details and maybe code are encouraged to verify the actual modification of this method over Maskgit.
>
> Thank you very much for the suggestion. We have added PyTorch code to clarify the implementation details (Figures A and B) and highlighted its difference from MaskGIT, which we believe greatly contribute to the paper's clarity. We appreciate any specific feed back on remaining confusions.
>
> ### [X-2] Class-conditional experiment added
>
> > This paper employs the unconditional MAGE as the baseline, lacking necessary verification on diverse models, e.g., the class-conditional Maskgit, and text-conditional model.
>
> We used the unconditional model to focus on distributional comparison aspects (rather than alignment required in conditional generation). However, we agree that comparison with conditional models would be valuable and have added experiments on class-conditional results with MaskGIT-PyTorch (Section E.2). We again confirm that the moment sampler closely approximates MaskGIT with the same temperature.
>
> ### [X-3] Performance gain and visualization
>
> > Even if on MAGE, the performance gain is marginal compared to the baseline as shown in Figure 3 and 4. Besides, a quantitative performance table are encouraged for better visualization.
>
> We basically do not try to surpass MaskGIT in image experiment; our aim in image experiment is to confirm that Moment approximates MaskGIT well, validating our theory (Figure 3). While this context is briefly described at the beginning of Section 5, we acknowledge that it is confusing, and we have added further explanation and Table A, which directly shows the difference between MaskGIT and Moment. We also added tables of actual FID values (Tables B, C). Regarding Figure 4, the limited improvement from caching is likely because MAGE/MaskGIT use a small number of tokens, and attention operations are not extremely heavy in this case.
>
> For language generation, there is indeed a significant performance gain. Our Hybrid method (and Hybrid+Cache) improves the quality-diversity tradeoff while achieving approximately 2× speedup in sampling efficiency.
>
> ### [X-4] Hardware dependency of partial caching
>
> > While the partial caching method aims for efficiency, the paper notes that its performance improvement is not significant on certain GPUs (e.g., H100 GPU), and computational overhead can even become noticeable (Section D.2.1). This may suggests that the practical benefits of this method might be highly dependent on hardware and model architecture, and its general applicability requires further investigation.
>
> The cause of this noticeable computational overhead is briefly discussed in Section D.2.1, but let us add further explanation.
>
> While the attention computation is rather limited in an intermediate step of partial caching, there are also increased operations beyond attention (such as token-wise forward passes and memory read/write operations for saving KV-cache). These are expected to scale sub-linearly with the number of tokens, but they include operations that are not as highly optimized as matrix computations. Therefore, on devices like the H100 where matrix operations are extremely fast, this advantage diminishes. Conversely, in cases where attention computation (which scales as $D^2$ with respect to the number of tokens $D$) becomes the bottleneck, the advantages of caching are more evident (as shown in the A6000 results).
>
> Partial caching, along with the exploration-exploitation tradeoff, should be viewed as one of the promising directions that can be leveraged within the choose-then-sample framework. Regardless of latency considerations, the ability to virtually increase the number of steps by examining only local Key-Value pairs has intrinsic value and potentially serves as a foundation for further caching methods.

---

### Official Review · Reviewer_DU35 · 2025-10-30

**Soundness:** 2
**Presentation:** 1
**Contribution:** 2
**Rating:** 4
**Confidence:** 3

**Summary:**

This paper investigates why MaskGIT’s parallel decoding sometimes produces inconsistent image quality. It reformulates MaskGIT sampling as a moment-based process, revealing that it implicitly performs temperature-scaled categorical sampling. Through this lens, the authors propose two improvements—partial caching and balancing—to stabilize token updates across steps. Partial caching reduces noise by reusing confident predictions, while balancing controls the trade-off between stability and diversity. Theoretical analysis and experiments show that combining both leads to higher-quality samples with fewer steps. These findings generalize beyond Moment to other parallel diffusion-style samplers like MaskGIT, showing the mechanism’s universality.

**Strengths:**

- There are too many notations, which makes the paper hard to follow, but if one manages to track them carefully, the explanations appear to be quite correct and well-founded.

- Naturally, in the MaskGIT sampler, increasing the strength of the added Gumbel noise would lead to sampling from a more smoothed distribution, and it was nice to see this explicitly stated at the beginning of Section 3.

- The difference from the original MaskGIT sampler is clearly demonstrated through Theorem 2.

**Weaknesses:**

- The paper defines too many unnecessary notations, which makes it difficult to read. For example, why did they even define $\beta=1+\frac{1}{\alpha}$? It seems that $\gamma$ is an important parameter, but it is only defined in the appendix through the algorithm, not in the main text.

- Honestly, it’s unclear whether the performance gap with MaskGIT is actually significant. In Figure 3, the temperature seems to be a more critical factor that influences performance. Would it be possible to show ImageNet samples generated by MaskGIT and the Moment sampler under the same random seed and number of steps?

**Questions:**

- In Figure 3, it seems that the optimal temperature varies across sampling steps. Can this phenomenon be explained mathematically?

- Are the methods introduced in Sections 4.1 and 4.2 applicable to the original MaskGIT sampler?

- Is the performance gain significant?

---

> ### Author Response · Authors · 2025-11-21
> **Rebuttal reply to Reviewer DU35**
>
> We appreciate your review, and let us respond to your questions and comments below:
>
> ### [D-1] Notation complexity
> > The paper defines too many unnecessary notations, which makes it difficult to read. For example, why did they even define $\beta=1+\frac{1}{\alpha}$? It seems that $\gamma$ is an important parameter, but it is only defined in the appendix through the algorithm, not in the main text.
>
> We initially defined $\beta$ because the form $1+1/\alpha$ appears repeatedly, which seemed redundant. However, we acknowledge that the relationship among $\alpha, \beta, \gamma$ was insufficiently explained. We have removed $\beta$ and added an explanation of $\gamma$ in the main text as highlighted in blue letters. We generally tried to follow the notation conventions from prior work, but we understand the readability concerns. We would highly appreciate any further specific feedback on notation.
>
> ### [D-2] We do not try to surpass MaskGIT with Moment
> > Honestly, it's unclear whether the performance gap with MaskGIT is actually significant. In Figure 3, the temperature seems to be a more critical factor that influences performance. Would it be possible to show ImageNet samples generated by MaskGIT and the Moment sampler under the same random seed and number of steps?
>
> > Is the performance gain significant?
>
> Figure 3 is specifically designed to verify our theory by demonstrating that Moment achieves performance equivalent to MaskGIT. While this context is briefly mentioned at the beginning of Section 5, we acknowledge it could be confusing and have clarified this by adding more explanation and Table A. While it is not about improvement, we have also added ImageNet samples by MaskGIT and Moment in Figure G.
>
>
> For language generation (where simply applying MaskGIT results in diversity collapse) in Section 5.2, there is indeed a significant performance gain. Our Hybrid method (that was enabled by MaskGIT -> Moment conversion) improves the quality-diversity tradeoff while achieving approximately 2× speedup in sampling efficiency (Figure 4, Right).
>
> ### [D-3] Optimal temperature varies across total steps
> > In Figure 3, it seems that the optimal temperature varies across sampling steps. Can this phenomenon be explained mathematically?
>
> This is a very important question. While it is difficult to rigorously predict which temperature is optimal, the transformation from MaskGIT to Moment sampler significantly advances our understanding. Since both methods achieve nearly the same performance as Temp, we can provide the following informal argument focusing only on token "sample" (excluding "choose" of positions):
>
> Consider unmasking $n$ tokens from a partially unmasked sequence. Comparing a single-step approach (unmasking $n$ tokens at once) versus a two-step approach (say, unmasking $n/2$ tokens per step), the latter is expected to have lower entropy in the distribution of the second $n/2$ tokens (i.e., more deterministic). This is because the additional conditioning from the $n/2$ tokens revealed in the first step would reduce stochasticity of the rest. Therefore, we can (informally) expect that increasing the number of steps leads to lower entropy. According to Proposition 3, without temperature sampling, this converges to the target distribution as we increase steps.
>
> However, with low-temperature sampling (i.e., with small $\alpha$), the convergence goes below the target entropy, since the sampling entropy is lower than non-temperature sampling. Therefore, there exists an optimal number of steps for each temperature (and conversely, an optimal temperature for each step count), and this optimal step count is expected to increase monotonically with $\alpha$.
>
> This understanding through sampling temperature is obtained by reformulating the problem using the Moment Sampler rather than the original MaskGIT framework.
>
> ### [D-4] Applicability to original MaskGIT sampler
> > Are the methods introduced in Sections 4.1 and 4.2 applicable to the original MaskGIT sampler?
>
> The improvements described in Section 4 are designed for the general choose-then-sample framework, and it would be difficult to directly apply them to the original MaskGIT sampler. This is because MaskGIT does not separate sample and choose (the choose operation depends on the probabilities of sampled tokens), making it impossible to modify "choose" independently. While it may be possible to emulate these methods in an ad-hoc manner for specific cases, this would likely require unnecessary resampling, leading to reduced efficiency.
>
> For example, to implement the Hybrid sampler using the MaskGIT approach without moment computation, one would need case-by-case handling: applying temperature sampling for sequences selected by Halton, and using conventional sampling elsewhere.
>
> The advantage of "Momentizing" MaskGIT is that, by completely separating choose and sample as in the Moment sampler, we gain flexibility to edit choose operations freely.

---

### Official Review · Reviewer_57uK · 2025-10-31

**Soundness:** 3
**Presentation:** 3
**Contribution:** 3
**Rating:** 4
**Confidence:** 3

**Summary:**

This paper proposes a novel sampling approach for Masked Diffusion models, termed Moment Sampler.
The authors provide a theoretical analysis of the MaskGIT sampler, which is commonly used in Masked Diffusion sampling, and introduce the Moment Sampler as an approximation to it.
They prove that under certain conditions, the proposed method well approximates the MaskGIT sampler.
Furthermore, they theoretically show that the loss of the generated samples obtained by the proposed algorithm is bounded in terms of exploration and exploitation, and to achieve this, they also propose an adaptive order selection method.
Experimental results demonstrate that the proposed method can effectively approximate the MaskGIT sampler across various domains, including image and language generation.

**Strengths:**

- The paper provides a strong theoretical analysis of the MaskGIT sampler.
This contributes to a deeper understanding of the mathematical foundation underlying how MaskGIT generates images.

- The authors propose the Moment Sampler, which effectively approximates the MaskGIT sampler while enabling the use of transformer caching, thereby reducing the computational complexity of the sampling process.

- The proposed approach is domain-agnostic and applicable to both image and text generation within the masked diffusion framework.

**Weaknesses:**

- The main limitation of the proposed method is that it does not surpass the performance of the MaskGIT sampler.
It is designed as an approximation method, and achieving a good approximation requires increasing the number of sampling steps (i.e., decreasing the demasking ratio).

- In Proposition 3, the paper claims that one-by-one sampling is unbiased; however, this theoretical advantage may not hold for generating large images or long sentences.
In practice, some degree of biased sampling may be unavoidable to maintain efficiency.

**Questions:**

- Error bound for the approximation in Eq. (2):
In Eq. (2), you state that the approximation holds when $N−k$ is large. Could you provide an error-bound analysis for this approximation? Specifically, for which ranges of $N−k$ does the approximation remain accurate, and within what quantitative error tolerance?

- In Appendix D, it is mentioned that partial caching requires more computation, but the reason for this is unclear.
What specific advantages does partial caching provide compared to the original MaskGIT sampler?

---

> ### Author Response · Authors · 2025-11-21
> **Rebuttal reply to Reviewer 57uK**
>
> Thank you for appreciating the strength of our theoretical contribution and generality of the work. Let us answer your questions and comments.
>
> ### [5-1] Moment does not surpass MaskGIT but leads to further improvement
>
> > The main limitation of the proposed method is that it does not surpass the performance of the MaskGIT sampler. It is designed as an approximation method, and achieving a good approximation requires increasing the number of sampling steps (i.e., decreasing the demasking ratio).
>
> While this observation is correct, increasing the number of sampling steps is empirically not needed for achieving good approximation; the theoretical guarantee is a worst-case bound. Moreover, the key advantage of the moment sampler is that it enables the decoupling of the sample and choose operations in MaskGIT, which opens up possibilities for improvements that are only feasible with the Moment sampler framework (as discussed from Section 4 onwards). Specifically, in the language experiment, through the progression from Moment -> U-Moment -> Hybrid, we prevent MaskGIT from destroying diversity, improve the trade-off, and actually reduce the time required to achieve the same perplexity by approximately half in terms of latency.
>
> ### [5-2] Unbiasedness of one-by-one sampling
>
> > In Proposition 3, the paper claims that one-by-one sampling is unbiased; however, this theoretical advantage may not hold for generating large images or long sentences. In practice, some degree of biased sampling may be unavoidable to maintain efficiency.
>
> The unbiasedness of one-by-one sampling is a theoretical demonstration that "choose" (index selection) does not add bias in the sampling distribution. What this implies is that under the choose-then-sample framework, performance generally tends to improve as the number of steps increases. Conversely, this does not necessarily hold for temperature sampling. Rather, we can infer that MaskGIT's performance deteriorates **because** it essentially employs temperature sampling (i.e., the problem lies in "sample", not "choose", as shown in Proposition 3).
>
> ### [5-3] Error bound for the approximation in Eq. (2)
>
> > In Eq. (2), you state that the approximation holds when N−k is large. Could you provide an error-bound analysis for this approximation? Specifically, for which ranges of N−k does the approximation remain accurate, and within what quantitative error tolerance?
>
> Thank you for the suggestion. We have added a concrete error bound and its derivation after Eq. (2). The estimate follows from Bernstein's inequality.
>
> ### [5-4] Computation of partial caching
>
> > In Appendix D, it is mentioned that partial caching requires more computation, but the reason for this is unclear. What specific advantages does partial caching provide compared to the original MaskGIT sampler?
>
> It mostly depends on how to compute the number of steps. We consider a full function evaluation as a step, so partial caching's "intermediate step" is just an additional computation to the first full-attention inference within a step. While the attention computation is rather limited in this intermediate step because of caching, there are also increased operations beyond attention (such as token-wise forward passes and memory read/write operations for saving KV-cache). These are expected to scale sub-linearly with the number of tokens, but they include operations that are not as highly optimized as matrix computations. Therefore, on devices like the H100 where matrix operations are extremely fast, this advantage diminishes. Conversely, in cases where attention computation (which scales as $D^2$ with respect to the number of tokens $D$) becomes the bottleneck, the advantages of caching are more evident (as shown in the A6000 results).
>
> Partial caching, along with the exploration-exploitation tradeoff, should be viewed as one of the promising directions that can be leveraged within the choose-then-sample framework. Regardless of latency considerations, the ability to virtually increase the number of steps by examining only local Key-Value pairs has intrinsic value and potentially serves as a foundation for further caching methods.

---

> > ### Comment · Reviewer_57uK · 2025-11-26
> >
> > Thank you for your response. Please find my follow-up comments below:
> > 1. Can the proposed Hybrid sampling resolve the loss of diversity that occurs during the MaskGIT sampling process? My interpretation of your explanation is that Moment sampling has issues with diversity, and you presented a solution to address that specific problem. Please let me know if my understanding is incorrect.
> > 2. To clarify, I was asking whether the $|J|=1$ setting is realistic. I am wondering if the process is unbiased under the assumption of demasking tokens one by one.
> > 3. Thank you for deriving the error bound. Based on the results, it appears to be influenced by $N-K+1$, $\epsilon$, and $|S|$. I am curious to know how this error manifests in the experimental results.
> > 4. I agree that caching is an important aspect. However, do the actual benefits gained from caching outweigh the performance of the MaskGIT sampler?

---

> ### Author Response · Authors · 2025-11-27
>
> Thank you for your follow-up comments. Let us answer the comments:
> 1. Your interpretation is correct, and from the fact that Moment sampler closely approximates MaskGIT, the Hybrid sampler also mitigate the diversity loss of MaskGIT. MaskGIT/Moment samplers have their implicit/explicit temperature sampling that makes generated text less diverse. By omitting the temperature sampling from Moment (i.e., introducing U-Moment), we can mitigate this issue. Then the Hybrid sampler further improves quality-diversity tradeoff. The actual output comparison between MaskGIT and Hybrid is given in Section E.5.2.
> 2. Actually, one-by-one unmasking or equivalent is widely used for masked diffusion, including LLaDA [1] and MDLM [2] etc. So it is realistic. It is also true that we need to effectively reduce the number of steps if we want to catch up with the efficiency of autoregressive models, which motivates our work, but we believe that many of the current pretrained diffusion language models use one-by-one sampling by default. Please let us know if this answers your question.
> 3. Since the error bound is an ultimate bound that holds for any marginal distributions $p_i$, we believe in practice the approximation is more accurate. Indeed, while the one-step ($k$ tokens at the same time) version of the derived bound (Theorem 2) only becomes tight when $N\gg k^2$, Moment closely approximates MaskGIT in almost all the experiments. This said, lookng at the experimental results (Figure 3(Left), Figure 5(Left)), we can observe that the error becomes slightly more significant when (1) the number of steps is smaller (i.e., $k$ is larger) and (2) the temperature $\alpha$ is smaller (i.e., $\lvert S\rvert^{1/\alpha}$ is larger), which generally aligns with the theoretical bound.
> 4. To clarify, caching is a technique that can also be applied to Moment (which shows an equivalent performance with MaskGIT), so we believe caching does not need to compete with MaskGIT. Rather, we are instersted in the trade-off between the performance gain (FID or Peplexity in the paper) and the additional computation of the intermidiate step of caching. In this regard, caching does not clearly improve the efficiency in image experiment (FIgure 4, Left), whereas it improves the efficiency in language experiment (Figure 4, Right). With an experiment with a larger text model (SDTT-large) we also observe this efficiency gain for 8-32 steps (Figure D, Right), so we believe our partial caching is worth trying when we want to improve the efficiency of diffusion language models.
>
> Thank you again for your response. Please let us know if these answers address your concerns or questions.
>
> References:
> - [1] Nie et al. Large language diffusion models. arXiv:2502.09992.
> - [2] Sahoo et al. Simple and effective masked diffusion language models. NeurIPS 2024.

---

> > ### Comment · Reviewer_57uK · 2025-11-27
> >
> > Thank you for your response. I have some follow-up questions regarding the points below.
> >
> > ---
> >
> > Regarding points 1 and 4:
> >
> > Is it correct to understand that the transition to CTS allows for the decoupling of temperature sampling, which in turn contributes to improved quality and diversity? Furthermore, does the shift to CTS enable caching, and does this caching imply approximating a longer sampling trajectory to enhance sampling quality?
> >
> > I would like to clarify the specific contributions of this paper. Is the problem statement defined as follows: by mathematically structuring the MaskGIT sampler, it becomes evident that MaskGIT relies on temperature sampling, which inherently compromises diversity? Therefore, is the core contribution that by converting this to CTS, you not only approximate MaskGIT but also resolve the diversity issues inherent in MaskGIT?
> >
> > ---
> >
> > Regarding point 2:
> >
> > To my knowledge, LLaDA involves a process of demasking multiple tokens and then re-noising them based on a specific probability. In such a sampling process, wouldn't it be possible for $|J| \neq 1$?
> >
> > My understanding is that $|J|=1$ implies demasking only one token per step. This is why I raised the question regarding whether this setting is realistic. If there is any misunderstanding on my part, I would appreciate your correction.

---

> ### Author Response · Authors · 2025-11-28
>
> Thank you for your timely response. Let us answer the follow-up questions:
>
> **Regarding points 1 and 4:**
>
> Yes, the reviewer's understanding is correct. Let us formalize it again in the following:
> - For caching, converting MaskGIT to its CTS version (Moment) allows us to apply caching, which virtually simulates a longer sampling trajectory with less attention computation.
> - The primary contribution of the paper can be summarized as follows. We mathematically restructured MaskGIT sampler in a choose-then-sample manner (Moment), which decouples the index selection and token sampling. It revealed that MaskGIT sampler exploited temperature sampling, thus harming diversity especially in text generation, and the decoupling enabled (1) fixing this temperature sampling to resolve the diversity issue and (2) improve the index selection part further by introducing Hybrid sampler and caching.
> - In a shorter sentence: **We mathematically coverted MaskGIT to a CTS equivalent with decoupled index selection and token sampling algorithms, which allows us to resolve the diversity issue and improve the index selection at the same time.**
>
>
> **Regaring point 2:**
>
> While the reviewer correctly describes the standard discrete diffusion sampler in theory, the implementation of LLaDA is actually different. If we look at their sampling algorithm (`generate()` in https://github.com/ML-GSAI/LLaDA/blob/main/generate.py), we can see that they precompute the number of tokens to unmask in `get_num_transfer_tokens()`. In their default setting of `steps==gen_length==block_length`, they just unmask one token per step. They concistently use this setting (e.g.,  see the table after "For the gen tasks of LLaDA-8B-Base, the evaluation result are as follows:" in https://github.com/ML-GSAI/LLaDA/blob/main/EVAL.md).
>
> To add more context, this kind of one-by-one sampling is also considered theoretically by [1] as "uniformization," exactly simulating the backward process of general discrete diffusion. In the case of masked diffusion, [2] proposes "first-hitting sampler" as a practical method, which is equivalent to one-by-one unmasking with a unifromly random index selection.
>
> References:
>
> - [1] Chen & Ying. Convergence analysis of discrete diffusion model: Exact implementation through uniformization. arXiv:2402.08095.
> - [2] Zheng et al. Masked diffusion models are secretly time-agnostic masked models and exploit inaccurate categorical sampling. ICLR 2025.

---

### Official Review · Reviewer_jaeH · 2025-11-01

**Soundness:** 2
**Presentation:** 2
**Contribution:** 2
**Rating:** 4
**Confidence:** 3

**Summary:**

This paper fouces on improving the sampling efficiency of masked diffusion models. The paper first provides a theoretical analysis of the MaskGIT sampler and show that it implictly incorportates a temperature-based sampling mechanism. Building on this insight, they propose the moment sampler, an asymtotically equivalent yet more interpretable choose-then-sampler approach. To improve efficiency, the paper introduces a partial cahcing technique and a hybrid strategy that formalize the exploration-exploitation trade-off during sampling.

**Strengths:**

* The paper provides a clear theoretical explanation of what conditoinal distribution the MaskGIT sampler draws samples from, effectively conneting it to the Gumbel-top-k tric. The analysis showing that MaskGIT sampling implicitly performs biased temperature-based sampling is insightful and could be valuable for future studies on masked diffusion model sampling.

* Building upon this analysis, the introduction of the moment sampler is reasonable. The proposed choose-then-sample algorithm is simple yet novel, enabling the integration of efficiency-enhancing technqiues discussed in the paper.

**Weaknesses:**

* While the theoretical analysis is convincing, the practical advantage of the proposed method as a more efficient sampler is not fully demonstrated.
  * The paper suggests that, from the experiments in language modeling, temperature sampling reduces generation diversity, but I want to know that the proposed approach mitigate this effect. Further empirical evidence and discussion would clarify this point.
  * Also, this language experiments appear to be conducted using only a single small pretrained model. Since the proposed method is described as a post-hoc sampling improvement, it would be valuable to verify whether the same effects and efficiency gains are observed across different models, including larger and more recent architectures.

* It could be beneficial to provide the derivation of the approximation in Eq. (2) in more detail. Since this approximation forms the foundation of subsequent formluation, a step-by-step explanation and, if possible, an error analysis would strengthen the rigor of the paper.

* The proposed method for handling the exploration-exploitation trade-off appears somewhat ad hoc. It is unclear whether simply selecting the first-m and -n elements from two orderings adequately captures the trade-off mechanism. Also, the proposed hybrid method seems asymmetric: swapping indices $i$ and $j$ may yield different $k$ even when the corresponding $m$ and $n$ are the same (e.g., $k=(4,3,2,6,5,1)$ in the paper's example). It would be helpful to discuss whether this asymmetry could cause any issues or biases.

**Questions:**

Please provide discussions or clarifications on the points raised in the Weaknesses section.

I think the theoretical analysis of the paper to be strong and insightful, but I would like to see more experimental evidence demonstrating the practical strengths of the proposed method. In particular, additional experiments across different models or settings that clearly highlight the efficiency and effectiveness of the proposed approach would significantly strenghten the paper.

---

> ### Author Response · Authors · 2025-11-21
> **Rebuttal reply to Reviewer jaeH (1/2)**
>
> Thank you for appreciating our theoretical contributions as "strong and insightful" and providing constructive feedback. Let us reply to your comments in the following:
>
> ### [j-1] Proposed approach mitigates diversity reduction from temperature sampling
>
> > The paper suggests that, from the experiments in language modeling, temperature sampling reduces generation diversity, but I want to know that the proposed approach mitigate this effect. Further empirical evidence and discussion would clarify this point.
>
> Yes, our proposed approach effectively mitigates the diversity reduction caused by temperature sampling. A key contribution of our work in this regard is that the moment sampler enables us to explicitly separate MaskGIT's index selection from its implicit temperature sampling mechanism. This separation reveals that temperature sampling is the primary cause of diversity reduction, not the adaptive index selection itself.
>
> In the language experiments, we demonstrate this through the progression from temperature-based to unbiased methods. Indeed, MaskGIT and Moment samplers are under 5.0 in terms of Entropy (Figure 5, Left), while their unbiased version U-Moment (and Hybrid) generally keeps the original Entropy range of 5.2-5.4 (Figure 5, Right). We have also added qualitative examples in Section E.5 (Figures H-K), which visually demonstrate how temperature sampling in MaskGIT leads to repetitive token usage (e.g., "I. and. and I." in Figure H, "the.A. and.A." in Figure J), severely harming diversity, while our Hybrid sampler produces more natural sentences with diverse vocabulary even with slightly higher generative perplexity.
>
> From a theoretical perspective, Proposition 3 states that choose-then-sample algorithms without temperature sampling ($\gamma=1$) converge to the correct distribution as we increase sampling steps, regardless of the specific positional selection strategy. This explains why unbiased methods can maintain proper entropy-perplexity tradeoffs, whereas temperature sampling breaks this convergence property.
>
> ### [j-2] Language experiments with different models
>
> > Also, this language experiments appear to be conducted using only a single small pretrained model. Since the proposed method is described as a post-hoc sampling improvement, it would be valuable to verify whether the same effects and efficiency gains are observed across different models, including larger and more recent architectures.
>
> > In particular, additional experiments across different models or settings that clearly highlight the efficiency and effectiveness of the proposed approach would significantly strenghten the paper.
>
> We appreciate this valuable suggestion and have added the following two language model experiments in Section E.3:
>
> - SDTT-large (863M parameters) [1]: We tested our methods on a significantly larger model compared to the original SDTT (169M parameters). Figure D shows that the Hybrid sampler consistently achieves better quality-diversity tradeoffs over Random, and both "Hybrid" and "Random+Cache" provide consistent speedups over Random across model scales. We used Llama3-8B for Gen. PPL computation instead of GPT-2 large so that SDTT-large is evaluated by a model larger than itself.
>
> - Di4C (modified architecture) [2]: We also evaluated our approach on Di4C, a model with a modified architecture that incorporates mixture modeling for capturing dimensional correlations (Hayakawa et al., 2025). Figure F shows that combining Hybrid with Di4C modeling (Di4C+Hybrid) achieves additive performance gains over SDTT, especially remarkable in the few-step regime, demonstrating that our post-hoc sampling improvements are complementary to architectural modifications.
>
> We believe these additional experiments strengthen the evidence that our methods indeed provide efficiency gains for different models without harming quality-diversity tradeoff.
>
> - [1] Deschenaux, J., & Gulcehre, C. (2025). Beyond autoregression: Fast LLMs via self-distillation through time. ICLR 2025.
> - [2] Hayakawa, S., Takida, Y., Imaizumi, M., Wakaki, H., & Mitsufuji, Y. (2025). Distillation of discrete diffusion through dimensional correlations. ICML 2025.
>
> ### [j-3] Detailed derivation of Eq. (2)
>
> > It could be beneficial to provide the derivation of the approximation in Eq. (2) in more detail. Since this approximation forms the foundation of subsequent formluation, a step-by-step explanation and, if possible, an error analysis would strengthen the rigor of the paper.
>
> Thank you for the suggestion and we have provided a detailed derivation along with error analysis in the revision (after Eq. (2)). The derivation is based on Bernstein's inequality.

---

> > ### Author Response · Authors · 2025-11-21
> > **Rebuttal reply to Reviewer jaeH (2/2)**
> >
> > ### [j-4] Asymmetry of Hybrid sampler
> >
> > > The proposed method for handling the exploration-exploitation trade-off appears somewhat ad hoc. It is unclear whether simply selecting the first-m and -n elements from two orderings adequately captures the trade-off mechanism. Also, the proposed hybrid method seems asymmetric: swapping indices $i$ and $j$ may yield different $k$ even when the corresponding $m$ and $n$ are the same (e.g., $k=(4,3,2,6,5,1)$ in the paper's example). It would be helpful to discuss whether this asymmetry could cause any issues or biases.
> >
> > We acknowledge that the merging strategy is heuristic in nature and indeed asymmetric regarding the ordering of two methods. However, the rankings from two methods naturally target opposing or independent objectives (exploration vs exploitation), resulting in minimal overlap in practice. We have conducted an additional experiment on Hybrid with swapped method ordering (s-Hybrid) in Section E.4, and observed a similar performance to the original Hybrid sampler, while s-Hybrid is slightly worse than Hybrid. This difference may come from the non-adaptive nature of Halton used in Hybrid, so we could potentially explore adaptive low-discrepancy methods such as determinantal point processes in future work.

---

> > > ### Comment · Reviewer_jaeH · 2025-11-28
> > >
> > > Thank you for the response. The authors' response have resolved several of my concerns, particularly those regarding the approximation error analysis.
> > >
> > > However, I still share a question similar to Reviewer 57uK regarding the role of CTS in the overall contribution. I would like to more clearly understand what the core contribution of the paper is. While the theoretical analysis establishing the approximation relationship between the MaskGIT sampler and the Moment sampler is compelling, it remains unclear to me how the introduction of CTS specifically addresses the limitations of temperature sampling used in existing methods.
> > >
> > > I would appreciate a clearer explanation of how CTS specifically resolves the shortcomings of temperature sampling, and how this connects to the theoretical and empirical contributions claimed in the paper.

---

> ### Author Response · Authors · 2025-11-28
>
> We are glad to hear that our rebuttal addressed some of the reviewer's concerns. We understand the remaining concern about the overall contribution. So let us answer the question more clearly.
>
> As we write in the new comment in the Reviewer 57uk's thread (added after Reviewer jaeH's comment), we can summarize our contribution as follows:
> **We mathematically coverted MaskGIT to a CTS equivalent with decoupled index selection and token sampling algorithms, which allows us to resolve the diversity issue and improve the index selection at the same time.**
>
> > how the introduction of CTS specifically addresses the limitations of temperature sampling used in existing methods.
>
> For this point, we would frame CTS as a flexible framework that can independently improve/modify index selection and individual token sampling at the same time. We can also opt in the temperature sampling, as is preferred in the MAGE experiment. Our contribution in this regard is that, by decoupling the index selection and token sampling of MaskGIT (which is competent in the image domain) to obtain Moment, we enabled its application to language generation by omitting the temperature of the sampling part (U-Moment). CTS allows further modifications of the "choose" part, which allowed us to establish the Hybrid sampler. The effectiveness of CTS is on the decoupling of "choose" and "sample," which are intertwined in the original MaskGIT sampler.
>
> > how CTS specifically resolves the shortcomings of temperature sampling, and how this connects to the theoretical and empirical contributions claimed in the paper.
>
> As we explain above, CTS is not solely for "resolving shortcomings of temperature sampling," since it can also adopt temperature sampling. This said, the diversity issue of MaskGIT with language can be solved by converting MaskGIT to Moment, a CTS sampler, since **being a CTS method allows the further transformations (`MaskGIT -> Moment -> U-Moment -> Hybrid`).** Let us track this sequence step by step. The corresponding theoretical result is bolded.
> - 1. The first transformation (`MaskGIT -> Moment`) is supported by our main result, **Theorem 2**. The approximation is also validated by our experiments with image and text (Figures 3, 5(Left), 6(Left) C; Tables A, D).
> - 2. The next (`Moment -> U-Moment`) is enabled because of the "decoupled" feature of CTS; we obrain `U-Moment` simply by disabling temperature sampling of `Moment`. Its effectiveness is partially supported by **Proposition 3** in that increasing steps does not lead to diversity collapse anymore.
> - 3. The final step (`U-Moment -> Hybrid`), where we hybridize the index selection of `U-Moment` with `Halton`. This is also made possible by CTS, because we can solely focus on the "choose" part unlike MaskGIT. (Actually, steps 2 & 3 are commutative since we edit "sample" and "choose" independently.) The effectiveness of this step is supported by our finding of exploration-exploitation tradeoff, presented in **Eq. (4)**. We can add `+Cache`, which is also enabled by the CTS transformation of MaskGIT.
> - Steps 2 & 3 (that are commutative) actually lead to empirical gains of efficiency without harming the quality-diversity tradeoff (Figures 4(Right), 5(Right), D-F).
>
> We believe it explains the role of CTS (decoupling of index selection and token sampling) and the meaning of converting MaskGIT to Moment. Please let us know if this explanation answers the reviewer's question.

---

### Author Response · Authors · 2025-11-21
**General response to Reviewers**

Dear Reviewers,

We are grateful to all the reviewers for valuable feedback and appreciating our theoretical contributions. While we answer individual questions in separate replies to each reviewer, let us summarize our revision of the paper here:

- **Section 3: Error bound for Eq. (2) and its derivation** (to Reviewers jaeH, 57uk): We have added a quantitative bound on the concentration inequality and its derivation after Eq. (2).
- **Section 3: Omitting $\beta$ and add explanation of $\gamma$** (to Reviewer Xexd): Regarding the notation, we omitted the non-necessary introduction of $\beta=1+1/\alpha$ and added more explanation of $\gamma$ at the end of page 4.
- **Section 5: Interpretation of image-generation results** (to Reviewers DU35, Xexd): As briefly described at the beginning of Section 5, we do not try to outperform MaskGIT by Moment in the image experiments in Section 5.1. Rather, we try to show that our Moment sampler can approximate MaskGIT well in image experiments. To mitigate the diversity collapse of MaskGIT in language experiments (Figure 5(Right), Figures H-K), deriving the Moment sampler has been essential to further lead to U-Moment and Hybrid in the language experiments in Section 5.2. Since our initial description in this regard could be confusing, we have added the "relative difference" between MaskGIT and Moment in Table A, and also added more explanations in the captions of Figure 3-5 to guide understanding the results.
- **Section A: Added PyTorch implementation of MaskGIT and Moment** (to Reviewer Xexd): To clarify the implementation of Moment sampler and its difference from MaskGIT, we added their sample PyTorch implementations in Figures A, B and highlighted the differences. We have also slightly modified Algorithm 1 (swapped Steps 1 and 2) to better align two implementations.
- **Section E.1 & E.2: FID results of MAGE and conditional MaskGIT** (to Reviewer Xexd): We have added detailed numbers of MAGE in Section E.1, and we conduct additional class-conditional ImageNet experiments with MaskGIT-PyTorch and added the results in Section E.2 to re-confirm the approximation ability of the Moment sampler.
- **Section E.3: Language experiments with larger/different models** (to Reviewer jaeH): We have added language experiments with two additional models: SDTT-large (863M parameters) and Di4C (modified architecture: mixture modeling). In both cases we confirm that our Hybrid sampler improves efficiency while keeping/improving quality-diversity tradeoff.
- **Section E.4: Asymmetry of Hybrid sampler** (to Reviewer jaeH): Since the Hybrid sampler is asymmetric with respect to the order of two methods in selecting indices, we have swapped the order of two methods (U-Moment and Halton) as an ablation study.
- **Section E.5: Generated samples** (to Reviewers jaeH, DU35): We added generated samples by MAGE (MaskGIT and Moment samplers) and SDTT (MaskGIT and Hybrid samplers) for qualitative comparison.

We again thank the constructive feedback from the reviewers, and we believe this revision greatly helps clarifying our contributions. Currently, we highlight the changes in blue and use alphabetic labels for tables and figures added to avoid changing existing numbering. We will rename them in the camera-ready version.

---

### Author Response · Authors · 2025-12-03
**Summary of discussion phase to ACs**

Dear Area Chair(s),

Since our discussion phase was closed before we conclude our discussions due to the information leak incident, we summarized our rebuttal and remaining discussion points to facilitate understanding. Please refer to the individual discussions with Reviewers for details.


Overall, the reviewers agree on the strength of the paper's theoretical contributions. We believe that Reviewer jaeH and 57uk (who responded during the discussion phase) agreed on most aspects of our rebuttal, with the primary remaining question being the clarification of our core contribution, whose discussion could not be fully concluded due to the incident. While the other two reviewers did not respond to our rebuttal, their primary concern appeared to be about our image experiments (i.e., "Is performance gain significant?"); this stemmed from confusion about the objective of image experiments, which we addressed by adding further explanations and tables. Let us summarize the reviewers' concerns and how our rebuttal/revision addressed them in the following.

### Concerns/points raised by more than one reviewers
- [j-1, 5-1] Core contribution of the paper
    - Q: What is the specific core contribution of this paper, specifically regarding the conversion to CTS (choose-then-sample)? Does it actually solve the diversity issue of MaskGIT?
    - A: It is the main concern that was remaining and shared by Reviewers jaeH & 57uk. As Reviewer 57uk also mostly correctly describes in their final reply, our core contribution beyond the theory is: **We mathematically coverted MaskGIT to a CTS equivalent with decoupled index selection and token sampling algorithms, which allows us to resolve the diversity issue and improve the index selection at the same time.**

        More concretely, this decoupling enabled (1) fixing the temperature sampling to resolve the diversity issue in the application of MaskGIT to text generation and (2) improve the index selection part further by introducing Hybrid sampler and caching. In contrast, MaskGIT's intertwined operations prevented such independent modifications. Finally, the role of CTS and its connction to our theoretical and empirical contributions is further explained in our final reply to Reviewer jaeH.
- [j-3, 5-3] Error bound for Eq. (2) and its derivation
    - Q: It is better to have an error-bound analysis for Eq. (2).
    - A: We added a concrete error bound using Bernstein's inequality, right after Eq. (2). This was acknowledge by both reviewers. Reviewer 57uk asked about how this error bound is reflected in the experiments, so we answered the following:
        >  lookng at the experimental results (Figure 3(Left), Figure 5(Left)), we can observe that the error becomes slightly more significant when (1) the number of steps is smaller (i.e., $k$ is larger) and (2) the temperature $\alpha$ is smaller (i.e., $\lvert S\rvert^{1/\alpha}$ is larger), which generally aligns with the theoretical bound.
- [5-4, X-4] Benefit of partial caching
    - Q: What are the practical benefits of partial caching, given its hardware dependency?
    - A: As we agree in the thread with Reviewer 57uk, the partial caching virtually increases sampling steps by examining only local Key-Value with reduced attention computation. While the non-attention overhead for this additional computation becomes noticeable on H100 GPUs (where matrix operations are extremely fast), it shows clear efficiency gains in the language experiments on A6000. It is true that the benefit of partial caching currently depends on hardware and individual experiments, but we regard it as one of the promising directions that was enabled by the CTS framework.
- [D-2, X-3] Interpretation of MAGE experiments and performance gain
    - Q: Why doesn't Moment surpass MaskGIT in image experiments? Is the performance gain significant?
    - A: Given that the MaskGIT sampler already works very well in the image domain, the image experiments (Section 5.1) are designed to validate our theory by demonstrating that Moment accurately approximates MaskGIT, not to outperform it (as stated at the beginning of Section 5). To mitigate possible confusions, we added Table A showing the relative difference between methods. The conversion from MaskGIT to Moment is essential for enabling subsequent improvements: in language experiments (Section 5.2), the progression `MaskGIT -> Moment -> U-Moment -> Hybrid` resolves MaskGIT's diversity collapse while achieving approximately 2× speedup, demonstrating significant practical gains. Finally, in response to Reviewer Xexd suggestion for better visualization, we added Tables B-D (with unconditional MAGE and class-conditional MaskGIT-PyTorch) in the appendix.

---

> ### Author Response · Authors · 2025-12-03
> **(Continued) Summary of discussion phase to ACs**
>
> ### Other points
> - [j-2] Language experiments with different models
>     - Q: More experiments including larger and more recent architectures?
>     - A: We added experiments with SDTT-large and Di4C (mixture modeling) in Section E.3, where we confirmed that the Hybrid sampler shows efficiency improvements in both experiments.
> - [j-4] Asymmetry of Hybrid sampler
>     - Q: Hybrid sampler is asymmetric about the ordering of two merged methods (e.g., U-Moment and Halton).
>     - A: We added an experiment of the order-swapped version in Section E.4. While it performs slightly worse than the original Hybrid sampler, both methods overall show similar performance.
> - [5-2] Unbiasedness of one-by-one sampling
>     - Q: Is one-by-one sampling discussed in Proposition 3 realistic? (Please see the first "Official Comment by Reviewer 57uK", point 2)
>     - A: It is realistic, as it is commonly used in practice, including LLaDA (please see our final reply to Reviewer 57uk for details).
> - [D-1] Notation complexity
>     - Q: Many unnecessary notations, including the introduction of $\beta$. Also, $\gamma$ is not explained in the main text.
>     - A: We replaced all $\beta$ with $1+1/\alpha$ in the main body. We also refined our explanation of $\gamma$ at the end of page 4. No answer to our question on other examples of unnecessary notations.
> - [D-3] Optimal temperature varies across total steps
>     - Q: Optimal temperature varyies across sampling steps in Figure 3. Why?
>     - A: We can heuristically understand this by Proposition 3. Please see our answer in "Rebuttal reply to Reviewer DU35."
> - [D-4] Applicability to original MaskGIT sampler
>     - Q: Can we apply methods in Section 4 to MaskGIT?
>     - A: No (in general), because they are for choose-then-sample methods. This also explains the advantage of converting MaskGIT to the Moment sampler, since we gain flexibility.
> - [X-1] Regarding token selection in "choose-then-sample" paradigm
>     - Q: It is unclear how tokens are sampled (in Algorithm 2). Adding code is encouraged.
>     - A: We added PyTorch implementations of MaskGIT and Moment samplers and highlighted their difference (Figures A & B in Section A).
> - [X-2] Class-conditional experiment added
>     - Q: Experiments with class-conditional MaskGIT or text-conditional models are missing.
>     - A: We added an experiment with a class-conditional model (MaskGIT-PyTorch) in Section E.2, and we again confirmed that the Moment sampler approximates MaskGIT well.

---

### Meta-Review · Area_Chair_agFZ · 2026-01-19

**Summary:**

The consensus was strong theoretical analysis of MaskGIT (Gumbel-top-k framing; “implicit temperature sampling” interpretation; and approximation to a CTS/Moment sampler), but reviewers mostly argue about borderline practical impact and clarity. We can see three recurring concerns: Unclear “core contribution” beyond theory specifically, what CTS conversion uniquely enables and whether it truly resolves MaskGIT’s diversity issue vs merely reframing the sampler. This was repeatedly raised and remained partially unresolved when discussion ended. Empirical significance is mixed and sometimes misaligned with expectations, for example image experiments did not show meaningful gains over MaskGIT (authors said they were meant as validation, not improvement), and multiple reviewers found the performance delta marginal / not compelling. Engineering contributions (Hybrid, partial caching) felt heuristic or hardware-dependent,  Hybrid’s merge rule looked ad hoc and asymmetric; partial caching benefits varied by GPU, raising doubts about general applicability. Despite improvements in the rebuttal (error bounds, code, extra experiments), no reviewer clearly shifted into an “accept”. That being said, a longer discussions could maybe clarify some points here.

**Reviewer Concerns:**

The rebuttal convincingly addressed several concrete, fixable concerns: it added a quantitative error bound/derivation for the key Eq. (2) approximation, clarified implementation details of the CTS/Moment sampler, and broadened empirical coverage with additional tables and extra experiments (including class-conditional image results and larger/different language models), plus an ablation for Hybrid’s asymmetry.

However, the most important concerns that remain outstanding are higher-level: reviewers (especially jaeH and 57uK) still found the core contribution and necessity of CTS somewhat unclear (i.e., whether CTS is truly essential versus a clean reframing to “turn off temperature”), the practical significance is still debated because image results do not show clear improvements over MaskGIT and the main gains are concentrated in specific language settings, and the key engineering components (Hybrid and partial caching) remain viewed as heuristic and/or hardware-dependent, leaving generality and robustness as lingering doubts.

**Reviewer Scores:**

Even in an optimistic scenario (let's say two reviewers move to 5), we'd get a mixed borderline profile. I guess we would still miss a clear accept advocate, it still tends to borderline reject.

---

### Decision · Program_Chairs · 2026-01-26

Reject